



# New experiments to probe the role of fractures in bedrock on river erosion rate and processes

Marion Fournereau[1], Laure Guerit[1], Philippe Steer[1], Jean-Jacques Kermarrec[1], Paul Leroy[2], Christophe Lanos[3], Hélène Hivert[4], Claire Astrié[1], Dimitri Lague[1]

[1]Univ Rennes, CNRS, Géosciences Rennes, UMR 6118, 35000, Rennes, France
   [2]Univ Rennes, CNRS, Lidar Platform, OSERen, UAR3343, Rennes, France
   [3]Univ Rennes, Laboratoire de Génie Civil et Génie Mécanique, Rennes, France
   [4]Univ Rennes, Inria, Géosciences Rennes - UMR 6118, IRMAR - UMR 6625, F-35000 Rennes, France

*Correspondence to*: Marion Fournereau (marion.fournereau@univ-rennes.fr) and Laure Guerit (laure.guerit@univ-rennes.fr)

**Abstract**

River erosion via abrasion and plucking is a fundamental process that impacts, among others, landscape evolution, sediment transport, and hillslope dynamics. It results from complex interactions between climate, tectonics, topography and the erodibility of bedrock. Despite its significant role, bedrock erodibility remains poorly understood as it is thought to aggregate

several parameters. Among these, fractures in bedrock are assumed to exert a strong control over erosion and thus, on landscape evolution. However, there is no systematic study of the impact of fracture geometry and density on bedrock river erosion. Due to the complex interactions at play, we investigate this question via an experimental approach. We develop an erosion mill apparatus designed to erode a fractured concrete disk with a diameter of 17 cm. We simulate fractures by embedding a 3D-printed plastic mesh in the concrete, using BVOH—a plastic that softens in cold water—creating mechanical

heterogeneities with a controlled pattern. We explore 10 different geometries and run 4 additional experiments without fractures for control. We record the topographic evolution every 2 minutes by photogrammetry and derive erosion maps by measuring elevation changes between successive scans. To our knowledge, this is the first study of its kind. Our results show that while fractures influence the relative contributions of abrasion and plucking, no clear relationship emerges between average erosion rates and fracture density or dip angle. However, we observe that the occurrence of plucking is related to the

density and the dip angles of fractures, and is favoured by intermediate density that scales with the size of the impactors, and intermediate dip angle that ease the removal of blocks. We suggest that the main impact of plucking is to change the location of erosion, increasing the eroded surface area rather than accelerating overall erosion rates. However, but as plucking accounts for at most one-third of the total erosion, its occurrence does not significantly affect average erosion rates. These findings emphasize the role of fractures on erosion mode and location rather than erosion rates, and highlight the need to further explore

the impact of fractures on riverbed erosion.

## 1 Introduction

Continental landscapes evolve through a combination of geomorphological processes, influenced by tectonics and climate Under inter-glacial conditions, river erosion acts as the primary mechanism responsible for the removal and transport of surface materials. The intricate interplay of various factors, including tectonics, climate, topography, and rock erodibility, contributes

to the complex dynamics of river erosion (Whipple, 2004; Whipple et al., 2022; Yanites, 2018). The erodibility of rocks is thought to be modulated by several parameters responsible for river incision, such as the mechanical characteristics of the rocks being eroded, and the size and supply of sediment (Bursztyn et al., 2015; Forte et al., 2016; Jansen et al., 2010; Sklar and Dietrich, 2001; Turowski et al., 2023b; Whipple and Tucker, 1999). However, the efficiency and relative importance of these different parameters in controlling erodibility are not well-known (Anderson and Anderson, 2010; Whipple and Tucker,

1999). Due to the numerous factors involved and the characteristic timescales of the erosion processes, experimental



approaches are ideal to address these questions (Paola et al., 2009). For instance, Sklar and Dietrich (2001) illustrated how sediments in transport and rock strength interact to control the efficiency of bedrock erosion. They demonstrate that erosion rate is maximum with a partial cover of the bed by sediments coarse enough to be transported as bedload. This partial sediment cover provides the tools for abrasion of the bedrock while maximizing the rate of sediment impact on the bed surface

(Chatanantavet and Parker, 2008; Lague, 2010). Rock properties also play a pivotal role in determining the resistance of bedrock to erosion. All else being equal, lithologies characterized by low tensile strength, such as sandstones or mudstones, tend to erode at higher rates than harder lithologies with higher tensile strength, such as granites or quartzites (Sklar and Dietrich, 2001; Turowski et al., 2023a, b; Zondervan et al., 2020). Lithology and rock fabrics can also influence the dominant erosion process, generally abrasion or plucking, even if cavitation or solution can also matter in some specific conditions (Scott

and Wohl, 2018; Whipple et al., 2000b).

Abrasion classically refers to the progressive wear induced by the impact of sediments on the bedrock substrate. This process occurs over extended time scales, creating structures such as ripples, flutes, and potholes, and is thought to lead to lower erosion rates compared to plucking (Anderson and Anderson, 2010; Beer and Lamb, 2021; Whipple et al., 2000b, 2022). In contrast, plucking entails the removal of blocks from the fractured bedrock substrate, leading potentially to localized higher

erosion rates over shorter time scales (Anderson and Anderson, 2010; Beer et al., 2017; Hurst et al., 2021; Scott and Wohl, 2018; Whipple et al., 2000a; Wilkinson et al., 2018). Molnar et al. (2007) and others argue that fracturing of rock by tectonics profoundly decreases their strength, and hence increases their rate of erosion, but also alters their mode of erosion by favouring the occurrence of plucking over abrasion where the bedrock has been fragmented into readily transportable sediments. The classical view is that abrasion dominates in pristine bedrock while plucking tends to prevail and lead to more rapid erosion in

highly fractured bedrock (Whipple et al., 2000b; Attal et al., 2006; Hurst et al., 2021; Lima et al., 2021; Scott and Wohl, 2018). Despite the general agreement over this double key role of fractures on erosion processes, few studies have systematically explored the links between fractures and erosion with quantitative approaches. In fact, despite these distinctions, accurately measuring and estimating the relative quantitative contribution of abrasion and plucking to the total erosion rate poses significant challenges (Beer et al., 2017; Whipple et al., 2022). In the following we do not discriminate between joints and

faults or between different fracture modes: all mechanical discontinuities, including potentially bedding planes and other geometrical features, are simply referred to as fractures (Eppes et al., 2024).

Recent studies on the role of fractures on erosion have mainly focused on hillslopes (DiBiase et al., 2018; Neely et al., 2019; Neely and DiBiase, 2020). For instance, DiBiase and Neely (2018, 2019, 2020) show that fracturing significantly influences rock erosion, sediment size, and slope of some hillslopes in California. They observe that hillslopes with higher fracture density

result in lower relief and smaller sediment blocks, while hillslopes with lower fracture density lead to more pronounced relief and larger blocks. This control of fractures on slopes also has an impact on erosion rates, as the most fractured hillslopes have erosion rates 2 to 5 times greater than less fractured ones (Neely et al., 2019). Regarding rivers, observations in the field indicate that rock fractures significantly impact landscape morphology (Colaianne et al., 2024), influencing both the mode and rate of erosion (Whipple et al., 2000b; Molnar et al., 2007; Scott and Wohl, 2018; Whipple et al., 2022). For instance, it is

generally observed that fractured bedrocks exhibit wide and rough river channels, while intact bedrocks display narrower and smoother channel features (Ehlen and Wohl, 2002; Scott and Wohl, 2018; Wohl, 2008). By analogy with rock drilling (Thuro, 1997), excavation (Pettifer and Fookes, 1994), or dredging (Vervoort and De Wit, 1997), it is suggested that in bedrocks with large fracture spacing with respect to impactors (i.e., sediments), plucking is not favoured and erosion rates seem independent of the presence of fractures (Molnar et al., 2007). Whipple et al. (2000b) also find that plucking dominates abrasion in natural

streams when bedrock is fractured over a submeter scale.

Moreover, it was suggested that a continuity of erosion processes operate at different scales, from wear abrasion (i.e., grain-by-grain abrasion) to macro-abrasion (i.e., block abrasion by chipping) (Whipple, 2004; Beer and Lamb, 2021). This is supported by laboratory experiments showing the scaling over 13 orders of magnitude of the abraded volume with the



impactor's kinetic energy, normalized by the tensile strength of the impacted bedrock (Beer and Lamb, 2021). Under both abrasion regimes, and assuming that fractures preferentially develop along pre-existing boundaries in bedrock, such as contacts between minerals, it is postulated that erosion is more efficient when the impactor size is similar to the size of the minerals constituting the bedrock, allowing to maximize the impact energy delivered to unit boundaries (Turowski et al., 2023b). Similarly, in most cases, plucking can only occur after a phase of fracture propagation to finish individualizing a pluckable block or to disintegrate an initial large block into smaller and more easily mobilized ones. This is favoured by hydraulic forces

(e.g., drag and lift, differential pressure between the block surface and its basal fracture), clast wedging and impact of coarse sediments (e.g., Whipple, 2000b). In addition to fracture propagation, the impact of coarse sediment also probably plays a large role in tilting and lifting the already-individualized block out of its initial position. The differences between macro-abrasion and plucking can therefore become scarce or confusing, as in both cases fractures play a key role and eroded fragments can be similarly large. Therefore, plucking is now generally defined as the mobilization of already fractured pieces of bedrock

under the action of hydraulic forces (Beer and Lamb, 2021) and could be extended to blocks mobilized during a sediment impact inducing limited fracturing or damage.

Based on this last definition, our study investigates how the geometry of already fractured rocks affects the magnitude, location and the dominant mode of erosion, by plucking or abrasion. We conduct laboratory experiments inspired by Sklar and Dietrich (2001), considering an additional variable: the fracture network geometry. Artificial concrete stones with different fracture

networks are eroded in mills with constant flow speed and sediment mass. We monitor the evolution of the topography at regular time intervals and derive erosion rates from topographic differences. We evaluate how the fracture geometry controls the capacity to erode by plucking compared to abrasion, and thus the local and short-term erosion rate shaping the topography.

## 2 Methods and materials

The experimental setup consists of an erosion mill system, inspired by previous designs (Sklar and Dietrich, 2001). It features

a Plexiglas column with a fractured concrete disk positioned at the bottom. Sediments and water are placed above the disk, while a rotating propeller generates sediment motion, driving the erosion process.

### 2.1 Experimental disks

Our fractured substrates are simulated with synthetic mesh placed in concrete disks. Each network is made up of two families of fractures with different spacing and dip angles (Fig. 1a). First, we perform experiments with square networks (i.e., the two

families have the same spacing) and vertical fractures. We use three different spacings (10, 20 and 40 mm) and repeat the experiments at least twice to account for intrinsic variability. Then, we perform experiments with rectangular networks (i.e., the two families have different spacings) and vertical fractures. We tested two configurations (10/20 and 10/35 mm) and repeated one of them (the 10/20 mm) to assess reproducibility. Finally, we perform experiments to explore the influence of the fracture dip angle, which represents the angle between the fracture plane and the horizontal plane. Starting from a square

network of fractures spaced by 20 mm, we change the dip angle of either one or the two families. We use two symmetrical networks (45/45 ° and 67/67 °) and repeat at least twice each experiment, and three asymmetrical networks (45/67 °, 90/45 ° and 90/67 °) that we did not repeat. In total, we used 10 different networks and ran 19 experiments with fracture networks. Different methods have been proposed to describe and characterise networks of fractures in natural rocks (e.g., Eppes et al, 2024). Here, the geometry is quite simple and we therefore only use two parameters: the fracture density, referred to as $p_{21}$

and defined as the total length of fractures over a given area (Dershowitz and Herda, 1992), and the sum of the fracture dip angles to distinguish the experiments with similar spacing but different dip angles. The experiments and their geometrical properties are summarized in Table 1.



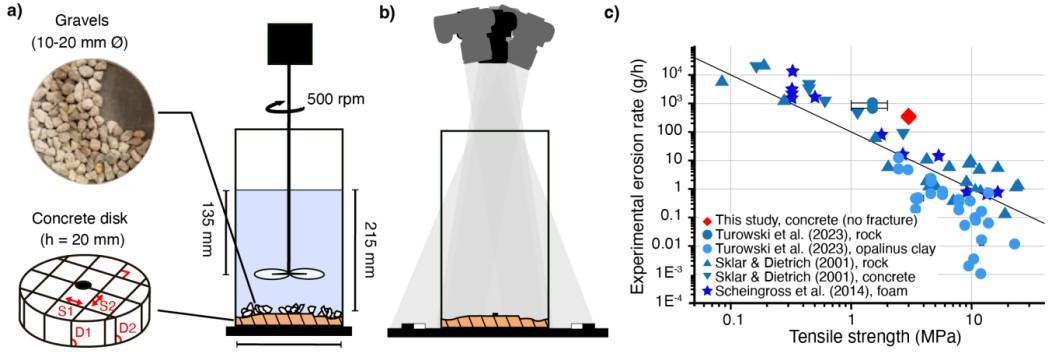

**Figure 1: General design of the experiments with a) the setup with the geometric parameters of the fracture networks that we explore in this study (with S and D the spacing and the dip angle of the two families of fractures, respectively), and b) the photogrammetry system. The concrete disk is fixed at the bottom of the column and erosion is launched by the motion of the water and gravels induced by the propeller. Every 2 minutes, the column is emptied and placed under 4 cameras. Targets are placed in the scene for absolute scale. c) Average erosion rate according to the tensile strength for various erosion mills (blue symbols) and for our concrete without fracture (red diamonds) (adapted from Turowski et al, 2023).**

We use OpenSCAD to model the fracture networks in 3D and a ZORTRAX M200 3D printer to print them, with a fracture width of $0.9 \pm 0.1$ mm. To create weak zones in the concrete disk without impeding its erosion, fractures networks are printed

using white BVOH (Butenediol Vinyl Alcohol Co-polymer), a thermoplastic that dissolves or becomes soft when in contact with cold water. Once printed, we use a circular mould (diameter = 172 mm, height = 20 mm) to pour concrete over the printed network. For experiments without fractures networks, the concrete is directly poured in the mould. We insert a 1.5 mm diameter plastic tube in the centre of the mould so that we can fix the disk on the column with a screw during the experiment. The concrete mix proportioning consists in one part of cement (CEM II/B-LL 32,5 N), three parts of Fontainebleau sand ($D_{50}$ =

210 µm and $D_{max}$ < 350 µm), and 15 % of water in weight. This composition was selected to ensure that the concrete disk erodes at a moderate rate—slow enough to track topographic evolution but fast enough to complete each experiment within a few hours. Concrete disks are removed from the mould after 3 days and left to harden for an additional 5 days in an airtight box. The disk is then slightly sanded to make it as flat as possible and fixed at the bottom of the Plexiglas column.

To characterize our concrete, we estimate the tensile $\sigma_t$ and compressive $\sigma_c$ strengths of our concrete at 8 days. Tests are

performed on prismatic samples referring to EN 196-1 (EN 196-1:2016 | May 2016, Methods of testing cement - Determination of strength) . We also estimate the mechanical properties of four additional concretes made with the same cement to sand ratio but with different water content (-20 %, -10 %, +10 % and +20 % with respect to the reference concrete). Despite these differences, the five concretes exhibit similar properties, with $\sigma_t$= 3.01±0.04 MPa and $\sigma_c$=10.11± 0.50 MPa. All experiments were made within this range of water content. For such mixes, the Féret relationships (Féret, 1892) relate their mechanical

properties to their composition:

$$\sigma_c = G \cdot \sigma_{ccj} \cdot \left(\frac{v_c}{v_c+v_w+v_a}\right)^2 = G \cdot \sigma_{ccj} \cdot (c)^2 \quad (1)$$

$$\sigma_t = G' \cdot \sigma_{ctj} \cdot \left(\frac{v_c}{v_c+v_w+v_a}\right) = G \cdot \sigma_{ctj} \cdot c \qquad (2)$$

with $v_c$, $v_w$ and $v_a$ the volume of cement, water and air per unit volume of concrete, respectively. $G$ and $G'$ act as granular coefficients. The compressive $\sigma_{ccj}$ and tensile $\sigma_{ctj}$ strengths at 8 days measured on the cement with standard sand (EN 196-

1) serve as indicators of the cement quality. For the selected cement, the values at 8 days, are $\sigma_{ccj} = 20$ MPa and $\sigma_{ctj} = 4.4$ MPa. Consequently, for Fontainebleau sand, $G = 4.9$ and $G' = 2.1$. The mechanical performances of mixes appear directly linked to the compacity $c$ of the paste (cement + water + entrapped air). While an increase in $v_w$ affects $c$, it also enhances the workability of the mix, reducing the amount of entrapped air. Interestingly, the reduction in air volume compensates for the



additional water, resulting in minimal changes to the mechanical strengths of the concrete within the studied range of water
content (Fig. S1).

## 2.2 Experimental protocol

For each experiment, the disk fixed at the bottom of the Plexiglas column is immersed in water until saturation of the connected porosity which occurs in about 20 minutes. To induce erosion, we add a constant volume of water together with granitic gravels of 10-20 mm in diameter on the top of the disk. To maximize erosion rates, the sediments cover about 2/3 of the surface of the

disk at rest (Sklar and Dietrich, 2001, Fig. 1a). Sediments are weighted at regular time intervals to ensure their mass remains constant. In only a few experiments, we added one grain during the run to keep the mass constant.

The rotation rate of the propeller placed in the column is 500 rpm so that the maximal flow speed at the edges of the column is of a few meters per second, and the sediment motion induces erosion by abrasion and plucking. Every 2 minutes, we stop the propeller, remove the water and the sediments, and take 4 pictures of the disk surface with fixed and remotely-triggered

cameras (Fig. 1b). An experiment ends when the bottom of the plexiglass column is reached, which corresponds on average to half a day per experiment and to about 60 minutes of effective erosion. For each experiment, we thus get 30 to 40 time steps. Such test duration appears sufficiently short to neglect the concrete strengths increases due to the continuous hardening of the cement.

We reconstruct the topography of the disks using photogrammetry with Agisoft Metashape, generating point clouds with

elevation (z) and horizontal coordinates (x, y). Since the four cameras remain fixed throughout the experiment, we use the 4D mode (i.e., a temporal series from fixed cameras), ensuring that the disks are consistently referenced in the same position.

To establish an absolute spatial scale, we place reference targets with known locations around the disk, which are automatically identified by the software. After reconstructing the topography as point clouds, we use the Canupo classification algorithm (Brodu and Lague, 2012) to filter out points classified as noise (mainly due to reflections on the Plexiglas tube), ensuring a

cleaner and more accurate dataset. . The remaining point clouds have a resolution of a few points per millimetre. Then, to obtain erosion maps from the topographies, we use the M3C2 algorithm (Lague et al., 2013) in vertical model to calculate the differences in elevation between successive pairs of point clouds. We use a regular grid of core points with one point per millimetre so that each cloud has about 21 000±1000 points. The projection scale in M3C2 is 2 mm, resulting on average to 25 points from each cloud used to compute the vertical difference.

To quantify the uncertainty associated with topographic reconstruction and point cloud differencing, we repeat 10 times this protocol with the same disk saturated with water (i.e., removing and replacing the column, taking pictures, generating the point cloud and 3D point cloud differencing). The maximum local difference in elevation between the 10 point clouds is 0.15 mm and we use this value as the topographic uncertainty in the following.

For practical reasons, we then convert the point clouds to rasters with 1 mm of resolution describing the elevation of the

topographic surfaces $z(x,y)$, the M3C2 differences $\Delta z(x,y)$ and the erosion rates $\Delta z(x,y)/\Delta t$, with $\Delta t$ the duration between two acquisitions (i.e., 2 minutes). In addition, at each time step, we weight the column after removing the water and the grains. These mass measurements only give a value averaged over the whole surface of the disk and are therefore less detailed than the topographic measurements. Therefore, they are only used to control that we accurately capture the erosion dynamics from the topographic evolution.

To compare our material to the ones from previous similar experiments (Turowski et al., 2023a), we run 4 experiments with no fractures and weight the column every 2 minutes to estimate the average erosion rates. Values are similar between the four experiments and match well with previous data (Fig. 1c). These experiments serve as a reference for the behaviour of our concrete without any fracture.



**2.3 Data analyses**

For each experiment, we derive two erosion rates from the erosion maps: i) the mean erosion rate, corresponding to the average erosion of the disk for each time interval and ii) the local erosion rate, corresponding to the erosion at each point of the raster and for each time interval. The mean erosion rate calculated from topographic differences correlates well with the weight measurements (Fig. S2). The average erosion rate for each experiment is defined as the mean of the mean erosion rates and we use one standard deviation as the associated uncertainty.

For each plucking event, we extract its location, area, volume (defined as the area times the change in elevation between the consecutive topographies) and time of occurrence, and we define the proportion (in %) of erosion by plucking with respect to total erosion as the ratio between the sum of erosion by plucking events during the whole experiment and total erosion.





**Figure 2: Temporal evolution of the topography and erosion rates of three different experiments. The first disk (a, b) has no fracture, the second (c, d) has a moderately dense network ($p_{21}$=50 m⁻¹, corresponding to vertical fractures with a spacing of 40 and 40 mm) and the third one (e, f) has a dense network ($p_{21}$=145 m⁻¹, corresponding to vertical fractures with a spacing of 10 and 20 mm). The time step between two pictures is 10 minutes.**

## 3 Results

In this section, we first explore the impact of fracture density by focusing on experiments with vertical fracture only. In the
second part of this section, we investigate the role of the fracture dip angle by focusing on experiments with fixed spacing
(20/20 mm) but variable dip angle.



### 3.1. The topographic evolution of experiments is linked to fracture densities

During an experiment, the topography of the disks evolves due to erosion, and we observe distinct patterns with and without fractures. In experiments with no fracture ($p_{21} = 0$ m$^{-1}$), we observe a continuous wear of the topography through time with a

clear radial pattern (Fig. 2a). At the end of a run, the topography is quite smooth and exhibits a radial symmetry. Indeed, the central part of the disk is barely eroded while about 1.5 cm of materiel is removed near the edge. This radial pattern is due to the rotative flow that induces a centrifuge force, pushing the grains toward the edge and generating a higher flow velocity and shear stress near the disk edge. Overall, this leads to a more frequent and more energetic impact of grains near the disk edge. In the experiment shown in Fig. 2a, the average erosion rate is 0.16±0.04 mm/min (Table 1) and is characterized by a strong

radial gradient, going from about 0 mm/min at the centre of the disk to about 0.6 mm/min on the edge (Fig. 2b). This behaviour (i.e., a smooth topography with radial symmetry and a radial pattern of erosion rates with limited intensity) is typical of experiments dominated by abrasion.

A similar behaviour is observed in experiments with a limited number of fractures ($p_{21} = 50$ m$^{-1}$). In such experiments, the influence of the fractures is visible in the topography (Fig. 2c) and we sometimes observe spots of high erosion rates located

near the fractures (Fig. 2d). They correspond to small plucking events but due to their limited size, they do not significantly modify the topographic evolution with respect to experiments with no fracture at all. The average erosion rate is slightly lower (0.14±0.03 mm/min, Table 1) but within the range of uncertainties.

On the contrary, experiments with a dense network ($p_{21} = 145$ m$^{-1}$) show a different topographic evolution. We still observe a radial pattern related to abrasion, with more erosion on the edges than in the centre (Fig. 2e). Yet, the topography is more

irregular with sharp changes in elevation. Erosion is no longer symmetrical nor regular through time, and we observe patches of high erosion rates, up to 1 mm/min, located all over the disk (Fig. 2f). They correspond to sudden removals of blocks (i.e., plucking events) and their size relates to the spacing of the fractures. The average erosion rate is only slightly higher than without fractures (0.17±0.05 mm/min, Table 1). In these experiments, abrasion and plucking coexist and lead to these specific patterns: an irregular topography with erosion over a large surface and superposition of locally high erosion rates on top of a

radial pattern of erosion. In all experiments, we observe that erosion occurs first on the edges of the disks before to progress inward (Figs. 2a, 2c, 2e).

### 3.2 Average and local erosion rates are barely controlled by fracture density or spacing

Having such a detailed temporal series of erosion maps is quite unusual, and we make use of this opportunity to investigate the temporal evolution of the mean erosion rate for each experiment. Here, we only focus on experiments with vertical fractures

and compare them against experiments with no fractures.

Despite obvious differences in topography and erosion dynamics, we observe a roughly similar temporal evolution of the mean cumulated erosion for all experiments (Fig. 3a). More variability is revealed when considering the mean erosion rate (Fig. 3b). During the first 10 to 20 minutes, the mean erosion rate tends to decrease before stabilizing between 0.1 and 0.2 mm/min (Fig. 3b, Table 1). Large variations are observed in experiments with fractures with mean rates up to 0.4 mm/min. These peaks in

mean erosion rate are associated with large plucking events occurring over the specific time step. For example, the sudden increase observed on the bold purple line around 30 minutes in Fig. 3a and 3b corresponds to the large plucking event visible in Fig. 2f. We observe also some variability in experiments without fractures, which suggests that abrasion shows some stochasticity. In addition, we observe no specific trend with respect to the fracture density on the cumulative erosion and the mean erosion rate (Fig. 3a and 3b). These results indicate that, in our experiments, the mean erosion rate does not discriminate

between experiments with or without fractures.





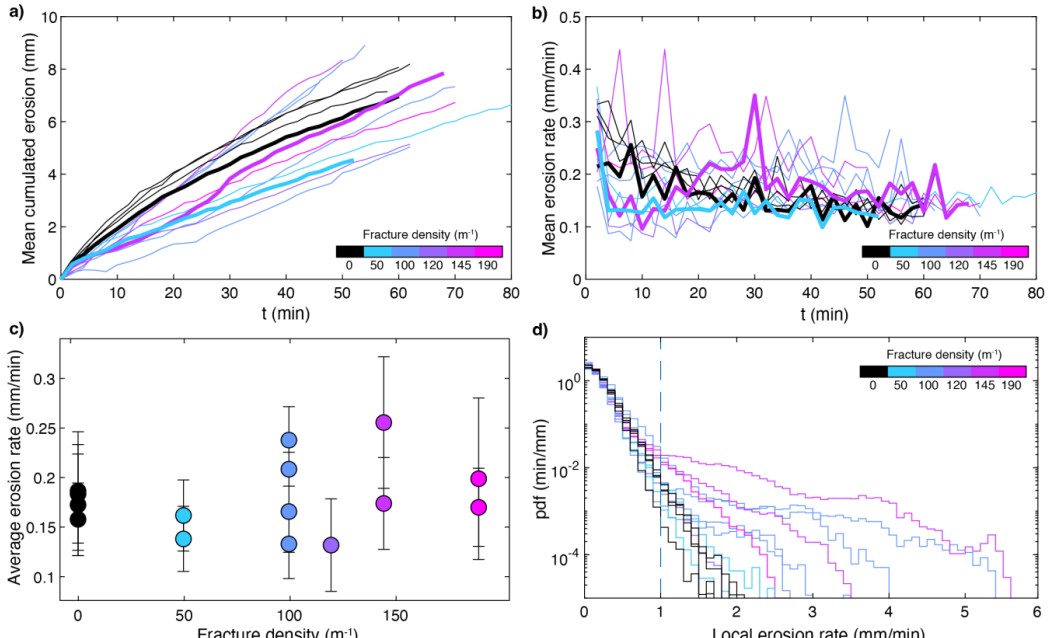

**Figure 3: Erosion through time of the different disks with: a) the cumulated erosion (in mm) and b) the mean erosion rate (mm/min). The bold curves correspond to the disks of Fig. 2. Distribution of c) the average erosion rates and d) the probability density function of the local erosion rates.**

Rather than looking at the experiments separately, we now group them according to their fracture density and look at the average erosion rates (calculated as the mean of the mean erosion rates shown in Fig. 3b). For the 11 experiments presented here, the average rates range from 0.13±0.05 mm/min ($p_{21}$ = 120 m$^{-1}$) to 0.26±0.07 mm/min ($p_{21}$ = 145 m$^{-1}$). We observe some variability between experiments performed with the same network, however it is within the standard deviations (Table 1) so that the variation in mean erosion rate during one run is larger than the variation between two similar experiments (Fig. 3c). The experiments with fracture density of 100 m$^{-1}$ and 145 m$^{-1}$ show larger spreads and higher average erosion rates (Fig. 3c). However, the range of values is similar to what is observed in other runs so that we do not observe a specific relationship between the fracture density and the average erosion rates (Fig. 3c, Table 1). As fracture density is related to the spacing of the fractures, these experiments show that the spacing of the fractures does not significantly affect the intensity of erosion.

Rather than looking at average erosion rates, we now explore the distributions of local erosion rates for all the experiments (Fig. 3d). All runs follow the same trend for local erosion rates of less than 1 mm/min, but two behaviours emerge at higher rates: the distributions of local erosion for the experiments with no fracture or low fracture density decrease exponentially while the distributions for experiments with high fracture density show a heavy-tail distribution (Fig. 3d). Such experiments are the ones prone to plucking (Figs. 2 and 3) and we therefore suggest that the shape of the local erosion rate distribution informs on the occurrence of plucking.

### 3.3 Occurrence and intensity of plucking is controlled by fracture spacing

For all experiments, we automatically detect the size and the timing of plucking events, defined as areas of 15 mm$^2$ minimum (based on visual inspection of plucking events from the time series of topographies) with an erosion rate of minimum 1 mm/min (based on the change in behaviour observed above 1 mm/min on Fig. 3d). As it is not possible to differentiate between one large event or multiple events occurring close to each other during the same time interval, in the following, one plucking event corresponds to the removal of one or several adjacent blocks during the 2-minute time interval.



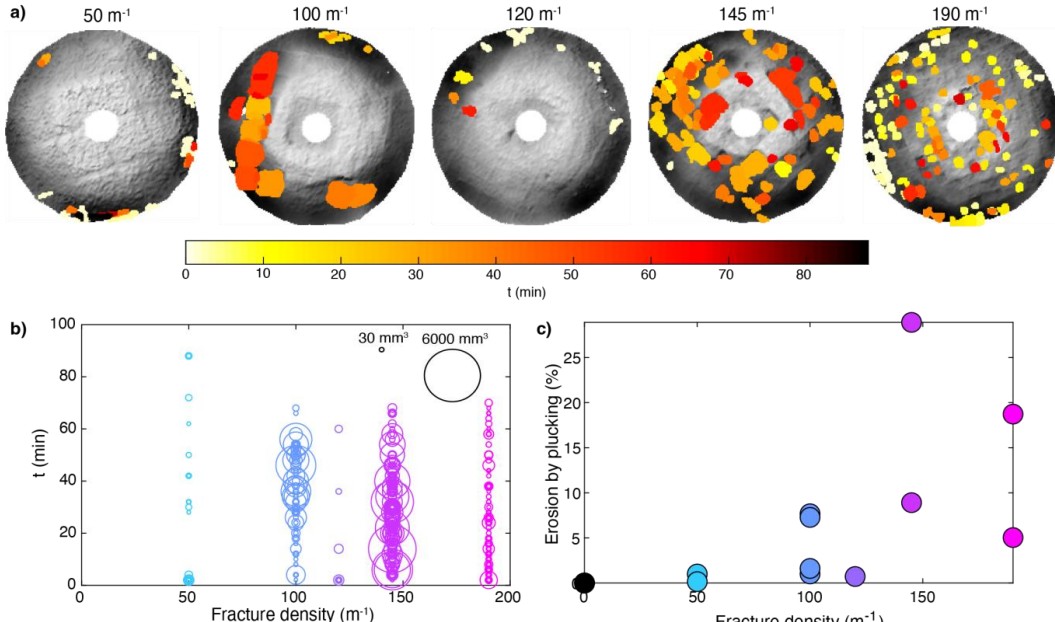

**Figure 4: Location, size and time of the detected plucking events for experiments with different fracture densities a) in map view (from 50m-1 to 190m-1 from left to right), ad b) as a function of time and fracture density for all experiments with vertical fractures. The size of the circles is proportional to the volume of the plucking event. c) Contribution of plucking to total erosion (in %) with respect to fracture density for experiments with vertical fractures.**

In all experiments, plucking events are observed over the whole duration of the run (Fig. 4a). When the fracture density is low, only a few plucking events located on the edges are observed whereas they tend to occur over the whole surface with higher

fracture density (Fig. 4a). In particular, experiments with a very dense network (190 m$^{-1}$) show numerous small plucking events with 90% of them smaller than 300 mm$^3$ and located all over the surface of the disk (Fig. 4a). The experiment with a fracture density of 120 m$^{-1}$ behaves as a low density one, and this could be related to the shape of the blocks (see Discussion).

Another way to document these patterns is to look at the time of occurrence and volume of events according to the fracture density (Fig. 4b). Plucking can occur at any time during a run and experiments with intermediate fracture density (≥100 and

<150 m$^{-1}$) have a higher tendency to remove large volumes by plucking than the other experiments. The absence of temporal trends supports the idea that our disks are quite easy to erode, with blocks already almost detached, as no major period of weakening is required before block removal can occur.

Finally, for each experiment, we calculate the proportion of erosion occurring by plucking with respect to the total erosion of the disk. We observe that the proportion of plucking to total erosion increases with increasing fracture density to a maximum

of 29 % in experiments with dense fracture network ($p_{21} = 145$ m$^{-1}$) and then seems to decrease for the highest density explored here ($p_{21} = 190$ m$^{-1}$) (Fig. 4c, Table 1). This decrease correlates well with previous observations that plucking events are numerous but not intense in the highest-density network experiments (Fig. 4a-b). This behaviour suggests that plucking is favoured in our experiments with an intermediate fracture density, corresponding to an average fracture spacing of 15-20 mm. Fracture spacing thus exerts a strong control on erosion mode by allowing plucking to occur. However, plucking is never the

dominant mode of erosion, as it accounts for a maximum of 1/3 of the total erosion (Fig. 4c, Table 1). In addition, due to the stochastic behaviour of plucking, experiments with similar fracture networks do not have the same contribution of plucking so we suggest fracture spacing only provides favourable conditions for plucking.




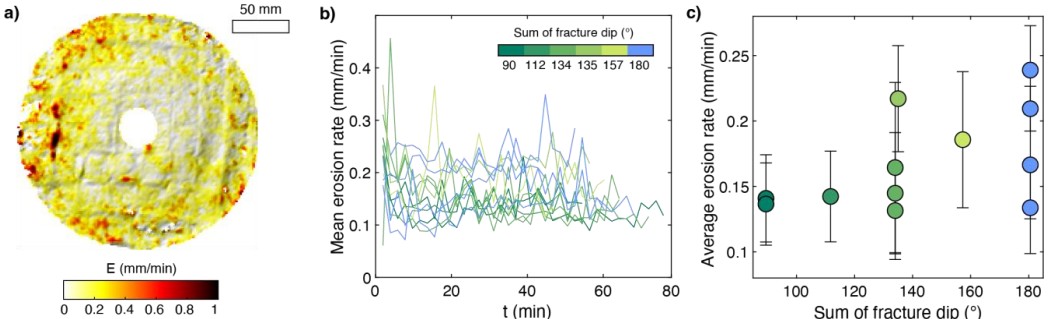

**Figure 5: Impact of the fracture dip angle on the average erosion rates with a) an example from a 45/45 ° experiment, b) the mean erosion rate with respect to time for the dip angle experiments, and c) the average erosion rate with respect to the sum of fracture dip angles. The bars correspond to the standard deviation, the blue colour corresponds to experiments with vertical fractures (90/90 °), and a spacing of 20/20 mm shown in previous figures.**

### 3.4 Fracture orientation with respect to the flow controls the occurrence of plucking

In natural environments, fractures are rarely all exposed vertically or normally to the riverbed surface. To further explore the role of fractures on erosion processes, we now focus on the seven experiments in which we vary the dip angle of the fractures together with the ones with vertical fractures and 20/20 mm spacing (Table 1). At first order, these experiments behave like the previous ones: 1) erosion occurs by abrasion and plucking, 2) erosion rates increase toward the edges (Fig. 5a), and 3) the mean erosion rates range between 0.1 and 0.3 mm/min and converge toward about 0.15 mm/min after 20 minutes (Fig. 5b). However, in details, our experiments show an impact of the fracture dip angle on erosion rates and plucking intensity and location. The average erosion rates range between 0.14±0.03 and 0.24±0.03 mm/min and are again associated with large standard deviations (Fig. 5c, Table 1). However, we note that experiments with large total dip angles (>135 °) display average erosion rates more scattered than the ones with low total dip angles (<135 °) and that the maximum average rates are observed from the higher dip angles (Fig. 5c). Plucking occurs in all runs and at any time during the experiment (Fig. 6a). The volume of plucked blocks is less spread than in the experiments with various spacing (Fig. 4b), supporting the idea that the size of the plucked blocks is correlated to the spacing between fractures. We note that the size of the plucking events is smaller for the experiments with low total dip angle than for the others (Fig. 6b). This suggests that when fractures are not vertical, it is difficult to remove large volumes by plucking due to the limited depth of a block.

Our results show that the higher the dip angle, the higher the proportion of plucking to total erosion (Fig. 6c). In fact, plucking is quite limited in experiments with low dip angles (45/45 °, 45/67 °) and its contribution to total erosion is of only a few percent (Fig. 6c). It increases to up to 10 % for intermediate dip angles (67/67 °, 90/45 °) and reach a maximum of 22 % for the 90/67 ° experiment. However, it decreases to less than 10 % when fractures are vertical (90/90 °) (Fig. 6c). We thus observe no clear relationship between the dip angle of fractures and we suggest that the main impact of fracture dip angle is the location of plucking events. In fact, in experiments with non-vertical fractures, we observe a preferential location of plucking events on one side of the disk (Fig. 6a), resulting in a quite dissymmetric pattern of erosion in such experiments. On the contrary, plucking events are more spread over the whole surface of the experiments with vertical networks (Fig. 4a) and erosion intensity is more homogeneous.



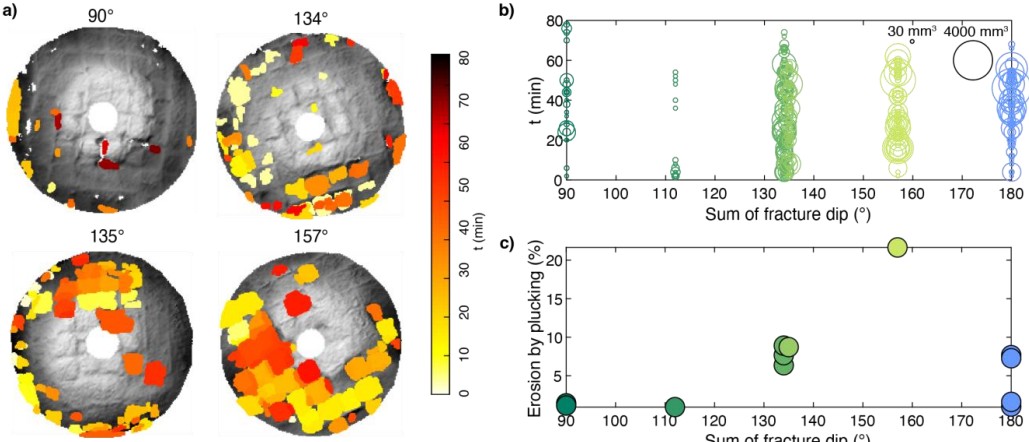

**Figure 6: Location, size and time of the detected plucking events for experiments with different fracture dip angles a) in map view (45/45 °, 67/67 °, 90/45 °, 90/67 °), and b) as a function of time and fracture density for all the experiments with different dip angles. The size of the circles is proportional to the area of the plucking event. c) Contribution of plucking to total erosion (in %) with respect to the fracture dip angles. The blue colour corresponds to the 90/90 ° network with a 20/20 mm spacing shown in previous figures.**

## 4 Discussion

In the discussion, we now consider all the experiments together. Our results demonstrate that fractures in bedrock exert an
influence on river erosion mode rather than on erosion rates. In fact, the average erosion rates of all our experiments vary within a factor 2 (from 0.13±0.05 to 0.26±0.07 mm/min, Table 1), but we could not identify a clear relationship with network geometry. We observe that the density of fractures controls the occurrence of plucking while the dip angle influences the location of plucking. In addition, the occurrence of plucking influences the location of subsequent plucking events. In consequence, the topography of the disk is controlled by fracture density with a smooth surface when there is no plucking and
a sharper one when it occurs. Plucking is never the dominant mode of erosion as even in the most favourable conditions, it accounts for less than 1/3 of the total erosion. Here, we explore some limitations of these results and propose some general interpretations.

### 4.1 Benefits and limitations of the new experimental setup with fractures

To our knowledge, this study is the first to explicitly integrate fractures in an experiment investigating river bedrock erosion.
This new experiment allowed us to investigate plucking as an emerging erosion process resulting from likely complex interactions between water flow, grain mobilization and impact, slip along fractures and damage in the concrete disk. To develop this new experiment, we relied on BVOH, a plastic material that dissolves when immersed in hot water, to print the synthetic fracture networks. However, in our setup, plastic dissolution is not fully achieved and softened plastic remains in the fracture plane of width 0.9 ±0.1 mm. As provided by the supplier, the intact plastic (i.e., before being in contact with water)
has a tensile strength of 45 MPa, which is greater than the one of the concrete (3.01±0.04 MPa). We can assume that the effective tensile strength of the softened plastic is less than its intact value, yet we have no clue whether it is lower than the one of the concrete. In case the tensile strength of the softened plastic is greater than 3.01±0.04 MPa, it could explain 1) why some plastic chunks were protruding of the disk surface in some experiments and 2) more importantly, why disks with fractures tend to erode at lower rates than the disks without fractures during the first 10-20 min (Fig. 3). Indeed, tensile strength is



considered as a suitable proxy for rock resistance to abrasion (Sklar and Dietrich, 2001, Turowski et al., 2023), and adding resisting plastic fractures to concrete disks should increase effective tensile strength and hence reduce erosion rates by abrasion. Moreover, the tensile strength of printed plastics tends to vary with the orientation. For instance, for a very similar plastic material (i.e., natural and not white BVOH), the tensile strength varies between 8.7 and 33.7 MPa, for the ZX (vertical plane) and XY (horizontal plane) orientation. If this information is not known for the white BVOH that we used in our experiment,

most printed plastics tend to be anisotropic (e.g., Grant et al., 2021). We therefore suspect that this likely dependency of tensile strength to orientation (i.e., larger tensile strength for fracture planes oriented horizontally than vertically) could partly explain why disks with fractures of limited dip angles have lower erosion rates and proportion of erosion by plucking (Figs. 5 and 6).

The synthetic fracture networks in our experiments also represent end-members in terms of fracture size distribution, as most
fractures have the same area or length, only varying due to the disk shape. In natural settings, fracture length tends to follow power-law distributions with negative exponents (e.g., Bonnet et al., 2001), leading to less frequent long fractures compared to small ones. At the first stage of the study design, we considered printing fracture networks based on more realistic size distributions using for instance a Discrete Fracture Network (DFN) model (e.g., Le Goc et al., 2019). However, out tests with DFN models lead to experimental issues such as isolated volumes which cannot be easily filled up by fresh concrete or isolated
fractures requiring numerous mechanical supports during printing.

To characterize the fracture network used in this study, we use the $p_{21}$ as a classic proxy for fracture density and the sum of fracture dips as a proxy for the vertical organisation of the fractures. However, these parameters do not fully describe the complexity of the networks as they account for only part of the geometry. For example, they give no information on the shape of the block formed by the fractures (square vs rectangle) or on the asymmetry of the fractures (45/45 ° vs 90/45 °). Yet it is
likely that such complexities play a role on the mode and location of erosion. We explored other parameters such as the average spacing of the fractures or the $p_{32}$, but we observed no or weak relationships with the average erosion rates or the erosion by plucking. We thus decided to keep the simplest proxies while keeping in mind that they are imperfect.

Therefore, we believe that future work should try to account for a more realistic fracture size distribution and a more accurate (yet simple) description of the fracture geometry to understand how this affects modes of erosion and the size distribution of
plucking events.

## 4.2 Geometry of plucking events

Our detection of plucking events is based on area and depth thresholds that we defined from visual inspection (area) and from local erosion distribution (depth, Fig. 3d) and applied to the difference between two consecutive topographies acquired at 2 min time interval. However, areas affected by plucking might also be eroded by abrasion during the 2 min time interval and
areas labelled as abrasion might have experienced small plucking events that are below our thresholds. In both cases, this would lead to a slight overestimation of erosion by either processes that we cannot quantify due to our limits to detect topographic changes (0.2 mm of uncertainties) and temporal resolution (2 min). In addition, these two mechanisms could be considered as a continuum of erosion processes (Beer and Lamb, 2021) so that using a sharp threshold, as we do here, imply that we do not capture this continuum.

Keeping in mind these limitations, we now look at the average depth of plucking events, defined as the plucked volume divided by the plucked area. Whatever the fracture density, the depth of plucking event, $d$, tends to increase with area, $a$, following a power law relationship in the form of $d = 1.5\ a^{0.15}$ (Fig. 7a). We suggest that the relationship between depth and area is related to the mechanical strength of the concrete. When plucking events are small, it is not possible to have small but deep plucking as it is mechanically difficult to dislodge from the surrounding rock mass. When plucking area increases, the depth
must increase as well to give the fragment some mechanical strength as large and thin ones are likely to break very easily and to detach as small plucking events. The maximal depth might be limited first by the capacity of the flow to detach concrete





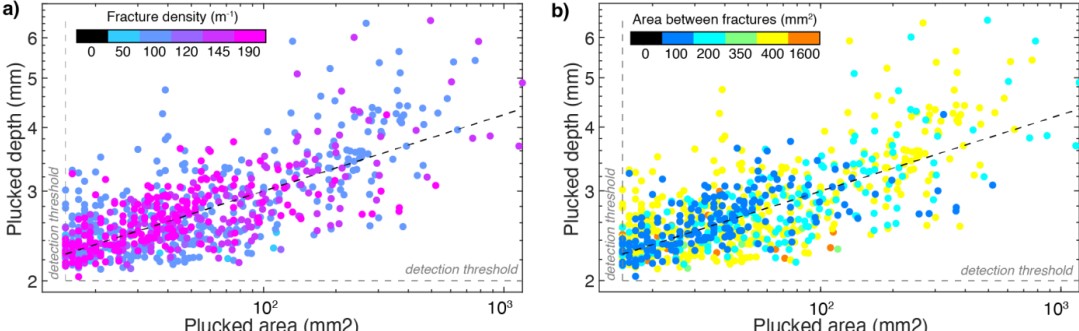

**Figure 7: Plucked depth with respect to the plucked area colored as a function of a) the fracture density and b) the area between fractures, for all the experiments. The grey dashed lines indicate the detection thresholds and the black one corresponds to the best fit through the data (see text for details).**

fragments and second, by the initial fragmentation of the concrete. In fact, all plucking events have a depth that is only part of the total disk thickness (20 mm), which implies that plucking occurs together with or after the sub-horizontal fracturing of the concrete. In experiments affected by plucking, we do not observe a time lag in the occurrence of plucking (i.e, plucking occurs

from the very beginning on the run, Figs. 4b and 6b). This suggests either 1) that the concrete is already partially fractured before the experiments start, which could result from the contraction occurring during the hardening phase, or 2) that concrete fracturing occurs rapidly under the action of water flow and sediment impacts during the experiment, due to its low tensile strength. We note that when a plucking event is detected, we are not able to differentiate between one single event or an amalgamation of several smaller events occurring next to each other or on the top of each other in the 2 min time interval. For

example, the large depths of 6 mm could simply be the removal of two 3 mm thick blocks at the same location. However, some of the concrete fragments we found when we emptied the column to photograph its surface are several mm thick (Fig. S3) supporting the idea that both mechanisms (unique or amalgamated event) are likely to occur in our experiments. The area of the initial blocks defined by the fracture network does not control the area of plucking events as we observe plucking events of any size in almost all experiments (Fig. 7b). However, high plucked depths are only observed in experiments with fracture

spacing of 20 mm (200 and 400 mm², corresponding to 10 by 20 and 20 by 20 mm, respectively). This suggests once again that this spacing is optimal to plucking. We note that it corresponds to the average size of the impactors used in this study (our gravels are 1-2 cm in diameter), but we would need dedicated experiments to further investigate this point. 4.3 Plucking and spatial patterns of erosion.

### 4.3 Plucking and spatial patterns of erosion

In the previous sections, we show that there is no clear relationship between the geometry of the fracture network (characterized either by the fracture density or by the total dip) and the average erosion rates (Figs. 3c and 5c). However, we observe a control of the fracture network on the occurrence of plucking, both in terms of plucking location (Figs. 4a and 6a) and of contribution to total erosion (Figs. 4c and 6c). In experiments with non-vertical fractures, we observe that plucking tends to be located on one side of the disks, inducing more intense local erosion (Fig. 6a). We suggest that this dissymmetric spatial distribution

relates to the orientation of the fractures with respect to the flow. Once plucking has been initiated, where the fractures face the flow (lower part of the bottom right disk on Fig. 6c for example), the local relief is sharper, which in turn promotes plucking. On the contrary, when the fractures are aligned with the flow (upper part of the bottom right disk on Fig. 6a), the local relief is smoother and impacts are less efficient, promoting abrasion. The tendency to pluck might thus not be directly related to the dip angle of the fractures but rather to interactions between the grains and the fractures that have the potential to

deviate the grains and to absorb part of the impact energy of the grains.



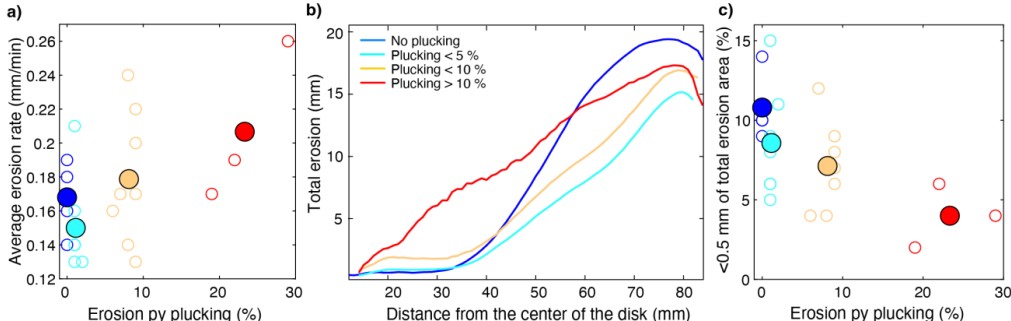

**Figure 8: Impact of the erosion by plucking on a) the average erosion rate, b) the total erosion after 40 min of run and according to the distance from the centre of the disk, and c) the area with limited total erosion, for all the experiments presented in this study grouped by their percentage of erosion by plucking (group 1, dark blue: no plucking, group 2, light blue: plucking <5 %, group 3, orange: plucking <10 % and group 4, red: plucking > 10 %). On panels a and c, white dots are for individual runs, coloured dots are for the average of the group. On panel b, only the average curves are shown.**

In an attempt to summarize all our observations, we now group the experiments according to the contribution of plucking to the total erosion of the disk (Table 1), whatever the geometry of the fracture network. Four groups emerge with no plucking (0 % of total erosion by plucking), low plucking (<5 %), limited plucking (>5 % and <10 %) and high plucking (>10 %).We calculated the average erosion rate of each group, and we observed that it increases with increasing erosion by plucking (Fig.

8a). In fact, the average erosion rate is close to 0.17 mm/min with no plucking and up to 0.21 mm/min with high plucking. In line with the results presented here, we note that the lowest rate (0.15 mm/min) is observed for experiments with low plucking (light blue in Fig. 8a) rather than for experiments with no plucking (dark blue in Fig. 8a).

To better understand this behaviour, we now look at the spatialization of erosion along the disks. For each experiment, we extract the radial profile of total erosion according to the distance from the centre of the disk after 40 minutes of erosion

(Supplement 4, Fig. S4). The profiles show quite large scatter and therefore, for each run, we determine the mean erosion profile, and for each group, we calculate the average cumulated erosion (Supplement 4, Fig. S4). In experiments with no plucking (dark blue, Fig. 8b), there is almost no erosion between 0 and 30 mm away from the centre of the disks. Erosion then increases with distance to about 70 mm before decreasing slightly toward the edges of the disk. These experiments with no plucking have the highest total erosion in the distal parts of the disk (distance >55 mm, Fig. 8b). In experiments with limited

plucking (light blue and orange, Fig. 8b), erosion is also very limited from 0 to 30 mm away from the centre of the disk and then increases radially, but slowly than for the previous group. Experiments with high plucking (red, Fig. 8b) have a similar evolution in the distal part of the disks (>40 mm from the centre) where the cumulated erosion increases continuously toward the edges. However, these experiments show a different pattern with a significant amount of erosion in the proximal part of the disk (between 0 and 40 mm) around 20 mm from the centre of the disk. Therefore, the main impact of plucking seems to

be the growth of the area submitted to erosion, extending towards the centre of the disk. To support this idea, for each group, we calculated the area that was only slightly eroded (defined as a pixel with less than 0.5 mm of total erosion over the whole duration of the experiment, based on the erosion profiles of Fig. 8b). We indeed observe that the proportion of low erosion areas decreases with the proportion of erosion by plucking (Fig. 8c).

In other words, the presence of fractures in the disks modifies the shape of the erosion profile and the location of plucking

events, leading to a counter-intuitive decrease in total erosion in experiments with low plucking and a limited increase for other experiments. The change in the spatial pattern of erosion seems to be the main control on the average erosion rates and suggests that plucking does not increase the erosion intensity itself, but rather extends the area submitted to erosion. Although plucking occurs from the very beginning of a run (Figs. 4b and 6b), the location of plucking tends to evolve through time as it is first mostly located on the edges of the disks before to progress toward the centre of the disks (Fig. 2). As more area is

eroded, the total erosion increases. This suggests that the main influence of fractures, whatever their geometry, is either to



absorb the impact energy or to deviate the grains so that there are fewer impacts on the edges of the disks, therefore less intense erosion but slightly more in the middle. This is beyond the scope of the present study to investigate these processes further.

**5 Conclusions**

In this study, we developed dedicated experiments of fractured concrete disk erosion to investigate how the geometry of
fractures in river bedrock can affect the magnitude, the location and the mode of erosion, by abrasion or plucking. We use temporal series of 3D topographies to document the erosion rates and patterns through time.

We show that when there is no fracture, the disks erode by abrasion only and show a smooth topography with a radial symmetry. On the contrary, when the disk is fractured, erosion can occur both by abrasion and plucking, leading to a rougher topography and less radial symmetry. However, we observe no clear relationship between the average erosion rates and the
fracture density or dip angle. This is partly because plucking never accounts for more than 1/3 of the total erosion in our experiments.

Rather than a direct impact on erosion rates, we suggest that the presence of fractures is a required condition for plucking to occur so that the first impact of fractures is on the morphology of the riverbed. Our results show that, in our setup, plucking is favoured by intermediate fracture density (145 $m^{-1}$), and by intermediate dip angle (67 °) which forms blocks that are thick
enough to be plucked more easily than with vertical fractures. In addition, we demonstrate that the orientation of the fractures with respect to the flow plays a major role in enhancing or not plucking. Finally, we show that plucking modifies the location of erosion so that more surface can be eroded. Therefore, we suggest that fracture density and dip angle, which can favour the occurrence of plucking, impact riverbed evolution by changing the mode and thus the location of erosion, rather than by promoting greater erosion rates.



This study highlights the need to consider the geometry of fractures in bedrock rivers to fully describe, understand and simulate erosion in rivers and channel evolution through space and time. To support the systematic integration of fractures in future works, a simple framework to characterize fracture density, shape and dip in bedrock rivers should be developed so that fracture geometry could become a classic measurement in fluvial studies. Finally, in line with previous works, the size of sediments with respect to the fracture geometry could be a key parameter that we intent to further investigate in the future, both in the

lab and in the field.

| n | Spacing (mm) | Density (m⁻¹) | | Dip angle (°) | Sum of dip angle (°) | | Mean erosion rate (mm/min) | Standard deviation (mm/min) | Erosion by plucking (%) |
|---|---|---|---|---|---|---|---|---|---|
| 1 | - | 0 | | - | - | | 0.18 | 0.05 | 0 |
| 2 | | | | | | | 0.19 | 0.06 | 0 |
| 3 | | | | | | | 0.16 | 0.04 | 0 |
| 4 | | | | | | | 0.17 | 0.05 | 0 |
| 5 | 40/40 | 50 | | 90/90 | 180 | | 0.16 | 0.04 | 1 |
| 6 | | | | | | | 0.14 | 0.03 | 1 |
| 7 | 20/20 | 100 | | 90/90 | 180 | | 0.24 | 0.03 | 8 |
| 8 | | | | | | | 0.21 | 0.02 | 1 |
| 9 | | | | | | | 0.17 | 0.04 | 7 |
| 10 | | | | | | | 0.13 | 0.03 | 2 |
| 11 | | | | 45/45 | 90 | | 0.14 | 0.03 | 1 |
| 12 | | | | | | | 0.14 | 0.03 | 1 |
| 13 | | | | 45/67 | 112 | | 0.14 | 0.03 | 1 |
| 14 | | | | 67/67 | 134 | | 0.16 | 0.06 | 6 |
| 15 | | | | | | | 0.14 | 0.05 | 8 |
| 16 | | | | | | | 0.13 | 0.04 | 9 |
| 17 | | | | 90/45 | 135 | | 0.22 | 0.04 | 9 |
| 18 | | | | 90/67 | 157 | | 0.19 | 0.05 | 22 |
| 19 | 10/35 | 120 | | 90/90 | 180 | | 0.13 | 0.05 | 1 |
| 20 | 10/20 | 145 | | 90/90 | 180 | | 0.17 | 0.05 | 9 |
| 21 | | | | | | | 0.26 | 0.07 | 29 |
| 22 | 10/10 | 190 | | 90/90 | 180 | | 0.20 | 0.08 | 9 |
| 23 | | | | | | | 0.17 | 0.04 | 19 |

**Table 1: Geometric properties, mean erosion rates with standard deviations and percentage of erosion by plucking for all the experiments. The colours to the right of density and sum of dip angle are the ones used in the figures.**

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
