# Peer review of "New experiments to probe the role of fractures in bedrock on river erosion rate and processes"

_EGUsphere, 2025_

## Referee Comment (RC1)

**Title:** New experiments to probe the role of fractures in bedrock on river erosion rate and processes

**Summary:**

Fournereau et al., performed a series of flume experiments to test the role of planar discontinuities in affecting the rates and spatial patterns of erosion in a bedrock-analog material (concrete). Planar discontinuities were produced from a network of BVOH plastic that dissolves or softens in cold water. The orientation and spacing of these materials inside the concrete was systematically adjusted, providing control on the spatial distribution and behavior of material properties of the eroded bed. An abrasion mill with rotating propeller was used to drive erosion with abrasive gravels. Erosion of the bed was measured both by weighing between 2 minutes of milling and by topographic differencing using 3D scans from close-range photogrammetry. The latter approach provides a map of where and how much erosion occurred on the bed over time. These topographic scans could be shared as digital data that records a time series of how erosion progressed through the bed material. This is a nice feature and produces a lasting dataset that could then be analyzed further.

The experiments show differences in the pattern and temporal evolution of erosion that occurred between concrete only and concrete + discontinuity experiments, but the overall amount of erosion remained similar. Removal of larger particles occurred when discontinuities were present and some of these fragments remained the flume cylinder, so their particle size distribution was measured after the experiments concluded. These fragments were interpreted as "plucked" blocks, as well as other areas in the topography where hot spots of erosion occurred. The relative proportion of erosion driven by plucked blocks and presumably detachment of smaller particles was estimated and shown to be a function of discontinuity spacing and in certain cases, orientation.

The experiments are rigorous and the hypothesis is tested appropriately. My main suggestions relate to the experimental apparatus and how the results are interpreted and presented with respect to field settings, which is challenging.

**Main suggestions:**

  (1)  *Experimental design with respect to plucking mechanism*

One potential issue with the experimental design -- combining the abrasion mill and a fractured bedrock media to quantify abrasion vs plucking erosion – is that it is not clear to what degree the mill produces flow conditions have been shown to drive plucking in fractured materials?

These flow conditions were generated and isolated in experiments with different flume geometries and shown to influence plucking of blocks:

Hydraulic jumps (Wilkenson et al., 2018) – the videos in the supplement are beautiful.

Flow constrictions (Saha et al., 2021) – a very large flume underlain by blocks

Knickpoints (Dubinski & Wohl, 2013) – also a larger flume underlain by blocks

Clemence 1988 – a student paper showing plucking at a knickpoint edge.

Protrusion/angle (Sidiki Koulibaly et al., 2024) – change block orientation & joint roughness

These are just suggestions of papers. They each include literature reviews of other works that should be considered in the topic.

Fractured bedrock tends towards these channel geometries (constrictions & knickpoints) in field settings due to heterogeneity in the rock strength fabric.  Not sure if a reference is needed here, but a 600 m long scan of a bedrock river section in Taiwan shows channel widths narrowing from >100m to 40m over a distance of a few 10s of meters for example (Carr et al., 2023).

The experiments showed in their topography scans that more variable topography emerges depending on the fracture network. The variable topography likely generates flow features listed above as the experiments progress. The self-emergence of these features seems like a significant finding to me that is clearly shown from the experiments, and a large portion of the paper is dedicated to describing and quantifying changing patterns of erosion that result.

It seems challenging to explore the significance of the non-steady flow fields in relation to plucking, given the mill apparatus design. It's difficult to interpret whether the experiment is representing these phenomena as they would be developed in a field setting. I'm curious what the authors think about digging into this point? The topography scans do exist...

The types of conclusions that can be drawn from the work are maybe a bit limited when applying to field settings if the above paragraph can't be quantified – namely it's tough to conclude a lack of increasing erosion effectiveness from adding fractures in a field setting? The self-emergence of a rougher bed seems like a significant point that could maybe be expanded on.

   (2) *Experimental design with respect to the 3D connectivity of the joint network*

The pre-seeded fractures do not contain an interconnected 3D network of joints. There are no subhorizontal fractures if I am understanding correctly? So blocks are not readily available for plucking – the 3$^{rd}$ joint set needs to form in-situ.

It is hard to quantify how the fracture network develops/grows in near surface stress fields, particularly quantifying the openness of 3D joint networks in valley bottoms, because the horizontal fractures would be mostly buried. Fracture surveys on pinnacles, spires, and waterfall edges seem to indicate that blocks are detached by 3D joint sets with open apertures though.

Flow underneath the blocks along a subhorizontal fracture set is important for generating lift or hydraulic jacking from sediments infiltrating cracks (Hancock et al., 1998). The latter mechanism would not be possible in this experiment given the aperture of the fractures (~1 mm) and the size of the sediment used (10-20 mm). This demonstrates scaling issues between the flume experiment design and particle size distributions, which could be significant for driving erosion in natural streams. In the field, the fracture aperture can be wider than a portion of the particle size distribution.

*(3) Strength of the joints?*

I am not sure how to interpret the material used to form the joints, but I do find it very interesting. Is the material harder or softer than the concrete, and how does it change over time? Could this be calibrated with submersion experiments without any sediment? In either case, there are landscape examples where joint material is softer (as in an aperture) or harder than the surrounding rock mass (joints infilled with carbonate cement or vein material – qtz for example), so there is significance in putting in the heterogeneous material with a pre-defined orientation & spacing in either direction.

These complexities (1-3) may limit the scope of interpretations that can be made from the experiments when moving to natural landscapes/channels, but the research question addressed by the experiment is very difficult.

The experiment is performed rigorously and with reliable measurements of erosion patterns. The experiment setup may apply to certain field settings and the general observation of more spatially variable/rougher bed topography created from the joints is important.

**General comment:**

I'd recommend if possible that the authors include some pictures of the bed as the experiments progress? My understanding is these abrasion mills turn into a mess when they are running, but some photographs of the disks (if fractures formed in-situ) or wear of

the BVOH might be insightful over time. Many flume studies include photographs or videos of the experiments progression, and I find this the be a big advantage to these experimental set ups.

**Line by line comments – many of these are minor grammar suggestions – sorry… these are annoying:**

Line 13: hillslope dynamics? Maybe "incision at the base of hillslopes" – if you have enough words to spare

Line 16: "However, there is no systematic study of the impact of fracture geometry and density on bedrock river erosion"

Slight wording changes suggested because this could be misinterpreted.

Maybe try:

*"Systematic studies comparing fracture geometry and density and bedrock river erosion are needed."*

Line 22: I'd remove "to our knowledge, this is the first study of its kind" --- if it's meant to shine, the work will shine on its own 😊

Lines 24-28 – any quantitative comparisons/statistics to report? In some of these lines (line 28) its not so clear whether this conclusion is based on the experiments or a general statement.

Line 29: Here is a place where it could be mentioned that erosion mode and location affects the roughness of the bed and presumably the flow field? Just a suggestion.

Line 40 – I like this point, but I would reword. Flume experiments also have their challenges upscaling to landscapes (which is usually a goal).

*"Experimental approaches can be an attractive approach to address these questions due to slow timescales of erosion and numerous conflating factors that could affect bedrock strength in field settings."*

Line 42: "demonstrated"

Line 43: "maximized"

Line 45: new paragraph?

Line 75 and others: unfortunately, the plural of "bedrock" is "bedrock".

Line 67-80 – In the context of fluvial erosion and bedrock fractures, there are fracture density measurements recorded in Snyder et al., 2003. I'll throw in a quick blurb here, but it's maybe not too important. In some ways DiBiase et al., 2018 or Neely & DiBiase, 2023 are making these comparisons between channel geometry, erosion, and fracture spacing, but the difficulty is that sediment size and bedrock fracture spacing co-vary in those sites (data from Neely & DiBiase, 2020), so the effect of one or the other is difficult to isolate – we interpreted that channels are mantled with sediment, so the fracture spacing effect is secondary to the cover effect. The flume approach is nice in this aspect -- the experiment can control the ratio between fracture spacing and sediment size in ways that might not be possible with field studies.

Line 81 – should plucking be included here? If the bedrock is already sectioned into blocks?

Figure 1 – I like this figure and it is nice to plot the control run in the context of the other studies with similar set ups.

Line 128: "center"

Line 170 – extra period.

Line 175 – interesting, this is a good idea.

Line 181, 185; in a few places in the manuscript would change "weighted" to "weighed"

Fig S2: interesting comparison, is the SfM of cumulative erosion always compared to the topography of the first point cloud? Or is erosion summed between successive differences as erosion proceeds? Just wondering if that could be a reason for the "drift" in cumulative erosion rates?

Figure 2 – Interesting to see the different spatial patterns in erosion and the emergence of the plucking holes

Line 199: "fractures"

Line 206: "eroded less" instead of "barely"

Line 212: in a propellor flume – not sure if there is a better word to describe the flume set up.

Figure 3 is very informative. I like panel d a lot. So if am reading this correctly, the largest local erosion rate is ~5 mm/min. Does this correspond to the size of a plucked block? Or are the blocks mostly wearing in place, in between the fractures? I guess with 10-20 mm impactors and 10-20 mm blocks, the impactors may not really be able to penetrate between the fractures?

Figure 4: Could it be more difficult to detect the small plucking events in the highest fracture density case?

I have a hard time understanding total dip angles? So two joint sets at 45 degrees would be 90 degrees?

Line 304: "Asymmetric"

Figure 5c, if more detail is needed to distinguish joint set orientations – i.e. same angle or different angle, consider using different symbols for the markers?

Figure 6 – I still have a bit of a hard time understanding the geometry by summing the dips together, maybe there is another way to display this. I think simply treating each combination of dips as a different column could be informative. Information about the fracture geometries gets lost by summing the fracture set dips together.

Line 319: I'm not sure, there are quite a bit of experiments that include fractures in the form of pluckable blocks? – maybe rephrase to be more specific about what is different about your experiment than some of the experiments listed earlier in the review.

Line 326: I really like the experiments, but it would be great to add some clarity here. Perhaps you can calibrate the material properties by submerging for regular intervals and measuring?

Line 328: "protruding from"

Line 331: "resistant"

Line 347: organization

Line 353: the geometries you are using seem like a good start to me?

Section 4.2 – I feel like something got disorganize here when explaining the methods of defining a "plucking event". Could this be elaborated on in the methods section 2.3? So moving part of section 3.3 for the methods into section 2.3? It seems unusual to me that a plucking event could be defined as eroding material that is smaller than the fracture-bound blocks, but this is maybe an outcome of not having a sub-horizontal fracture set as well?

I see the fragment sizes plotted as a figure in the supplement. This is pretty interesting and does show more "platey" fragment sizes. Do the fragments cluster around the fracture spacing dimensions? I would expect the clasts corresponding to the densest fracture density to produce a cluster of fragments near 10mm a-axis, 10 mm b-axis, if the partings were occurring along the fractures?

I guess I am trying to untangle what went on inside the flume during detachment. The fragments are helpful but as you say they are an indirect measure. Glad you measured and held onto them!

Line 363: "implies"

Figure 7: Interesting plots that are getting at some of the geometry questions from before. So 10x10 mm is the spacing between the densest fracture network, but can produce plucked blocks that have a z-axis between 2.2 and 4 mm? The units of this are a bit hard to follow with the dimensionality of a and d, and the fit power law form.

Maybe plot without log transformation? If you are producing cubes, the relationship should be a square root form? So this is implying that slabs are formed (makes sense for plucking)?

Line 370: "ones?" fragments?

Discussion section – there is limited discussion linking the results to past work on the topic – either other flume experiments or contextualizing the results with bedrock erosion in the field. Maybe this can be developed a bit more?

It is important to discuss the data and the specifics of the experiment & design, but a section that zooms out to contextualize the study with past work/field settings would be encouraged – (by me)

Line 436: remove "the" before erosion rates

Line 437: "no fractures"

Line 442: I like this sentence for the implications of the work.

Line 446: remove "or not"

Line 448-449: I would be careful about the last phrase, unless addressing the experimental design/questions earlier in the review.

Thank you for sharing your interesting work and it was a pleasure to read 😊

If you would like to discuss more or if you have any questions about this review, please do not hesitate to contact (abn5031@arizona.edu)

Referenced studies:

Carr, J.C., DiBiase, R.A., Yeh, E.C., Fisher, D.M. and Kirby, E., 2023. Rock properties and sediment caliber govern bedrock river morphology across the Taiwan Central Range. *Science Advances*, 9(46), p.eadg6794. https://doi.org/10.1126/sciadv.adg6794

Clemence, K.T., 1988. 1987 Student Professional Paper: Undergraduate Division: Influence of Stratigraphy and Structure on Knickpoint Erosion. *Bulletin of the Association of Engineering Geologists*, 25(1), pp.11-15. https://doi.org/10.2113/gseegeosci.xxv.1.11

Dubinski, I.M. and Wohl, E., 2013. Relationships between block quarrying, bed shear stress, and stream power: A physical model of block quarrying of a jointed bedrock channel. *Geomorphology*, 180, pp.66-81. https://doi.org/10.1016/j.geomorph.2012.09.007

Hancock, G.S., Anderson, R.S., and Whipple, K.X., 1998, Beyond power: Bedrock river incision process and form: in Tinkler, K., and Wohl, E. E., eds., Rivers over rock: Fluvial processes in bedrock channels : Washington, D.C., American Geophysical Union, Geophysical Monograph 107, p. 35-60.

Koulibaly, A.S., Saeidi, A., Rouleau, A. and Quirion, M., 2024. Laboratory physical model for studying hydraulic erodibility of fractured rock mass. In *New Challenges in Rock Mechanics and Rock Engineering* (pp. 1416-1422). CRC Press.

Saha, R., Lee, J.S. and Hong, S.H., 2021. The impact of lateral flow contraction on the rock plucking process under sub-critical flow conditions. *Earth Surface Processes and Landforms*, 46(14), pp.2902-2915. https://doi.org/10.1002/esp.5220

Snyder, N.P., Whipple, K.X., Tucker, G.E. and Merritts, D.J., 2003. Channel response to tectonic forcing: field analysis of stream morphology and hydrology in the Mendocino triple junction region, northern California. *Geomorphology*, 53(1-2), pp.97-127. https://doi.org/10.1016/S0169-555X(02)00349-5

Wilkinson, C., Harbor, D.J., Helgans, E. and Kuehner, J.P., 2018. Plucking phenomena in nonuniform flow. *Geosphere*, 14(5), pp.2157-2170. https://doi.org/10.1130/GES01623.1

**Esurf review aspects:**

1.  Does the paper address relevant scientific questions within the scope of ESurf?

    Yes

2.  Does the paper present novel concepts, ideas, tools, or data?

    Yes

3.  Are substantial conclusions reached?

    Yes

4.  Are the scientific methods and assumptions valid and clearly outlined?

    Yes

5.  Are the results sufficient to support the interpretations and conclusions?

    Not necessarily – see review.

6.  Is the description of experiments and calculations sufficiently complete and precise to allow their reproduction by fellow scientists (traceability of results)?

    I think so?

7.  Do the authors give proper credit to related work and clearly indicate their own new/original contribution?

    Not necessarily – see review.

8.  Does the title clearly reflect the contents of the paper?

    Yes

9.  Does the abstract provide a concise and complete summary?

    Yes

10. Is the overall presentation well structured and clear?

    Yes

11. Is the language fluent and precise?

    Yes

12. Are mathematical formulae, symbols, abbreviations, and units correctly defined and used?

Yes

13. Should any parts of the paper (text, formulae, figures, tables) be clarified, reduced, combined, or eliminated?

Yes

14. Are the number and quality of references appropriate?

Not necessarily – see review.

15. Is the amount and quality of supplementary material appropriate?

Yes

---

## Referee Comment (RC2)

I generally agree with the review provided by the other referee on this manuscript. I think that they summarized the experimental setups and findings well. This is an interesting body of work, and I think that my main concerns lie with how the experimental set-up affects the ability to generate substantial plucking and how to relate this to a field setting. I also have concerns in the definition of plucking as used here, which could be a misconception on my part of the experimental setup. In any case, this work shows interesting relationships on how fracture spacing generates autogenic roughness in a bedrock channel. Additionally, I recommend that the authors read throughout their manuscript for spelling and grammar errors and casual writing. An example is line 326 "yet we have no clue whether it is lower…"

**Experimental setup:**
I had a hard time understanding the fracture network within your discs. Were the fracture networks 3D, or did you just have 2D fractures? I think that a conceptual figure for both this and how you sum up dip angles and fracture lengths would be really helpful.

With an abrasion mill type setup, you aren't really able to replicate the processes involved in plucking. With a purely abrasion scenario, grains circulate, eroding the bed to either produce more grains or finer material. In nature, plucking is generally represented as knick progression upstream, a sort of unraveling of the bed. In addition to a fracture network of sufficient density to create blocks that are erodible under realistic hydraulic conditions, this requires a downstream boundary of an exposed block. This can occur due to vertical lift, base level drop, extreme abrasion, etc. In the abrasion mill setup, you don't have the ability to initiate the plucking with a downstream boundary (at least as you have set it up with a flat bed). Which in and of itself is okay, because you just have to simulate the 'first' plucking event by lift to trigger plucking. But then the amount of erosion by plucking that you are able to generate is only one circumference of your circle until you reach the first block again. Then you must re-create the higher erosional threshold first plucking event to trigger a wave around your disk. Further, in the real world these larger blocks would wash downstream and exit the system (or deposit downstream). What happens to them in your mill? This is not necessarily unrealistic, but deserves discussion. I think also this hinges on your blocks being fully separated from the bed and readily mobilized. If you don't have a 3D fracture network, then are these events really plucking? Or are sediment impacts able to abrade larger chunks of the cement between fractures?

Due to the circular nature of your disk, the fracture orientation relative to flow is never consistent so it is difficult to draw conclusions about fracture orientation and dip angle. These orientations relative to the flow direction are extremely important, and having them change is likely a cause of the variability in your data.

Line 254-255 "and we therefore suggest that the shape of the local erosion rate distribution informs on the occurrence of plucking." Considering that plucking contributes to the erosion rate distribution, the erosion rate distribution cannot control the occurrence of plucking. Maybe this needs to be reworded for clarity.

What are dimensions of blocks created by your fracture network? Block geometry has been shown to have a significant impact on plucking mode and erodibility (see Lamb et al., 2015; Hurst et al., 2021; Lamb and Dietrich, 2009), and so the dimensions of blocks created by your fracture network should affect how 'pluckable' the blocks are.

It would be interesting to run these experiments with a range of sediment inputs. I'm curious if with your setup you would even get any 'plucking' if you don't have sediment impacts since it seems you don't have fractures at the base. The total percentage of erosion by plucking will depend on how abradable your bed is compared to how pluckable your blocks are.

Why do you think less plucking in the highest fracture density? Are the plucked blocks so small that they are below your threshold of detectability for plucking?

Fig 5/dip angle variability conclusions. How many runs did you have at each dip angle? It looks like you had greater variability in runs where you conducted more repetitions. So that isn't a strong conclusion if you only had 1-2 runs in the lower variability cases and would need further discussion and analysis.

**Discussion:**
While your discussion section does a nice job of talking about all of your experimental runs together, I think that more needs to be done to put your work into the context of previous work and discuss how this applies to natural settings.

In discussion and throughout paper you need to be clear that you are exploring a limited scenario where you have a constant supply of sediment. This interaction of sediment can really impact the dynamics of plucking vs. abrasion. You also have a highly erodible bed that is susceptible to abrasion, which can influence the dominance of abrasion over plucking in terms of overall erosion. To me, the fact that there is so little difference in average erosion rates indicates one of two things. Either a) the experimental setup is preventing a greater magnitude of plucking from occurring and since the bed can't unravel, the stochastic plucking events are contributing a great magnitude of erosion that is averaged out over time (a real thing that happens where rapid events with high magnitudes of erosion are interspersed with long periods of no erosion!!! i.e. jokulhaups in Iceland. So a cool interpretation if you can back it up with observations or data) or b) erosion by abrasion dominates the long-term erosion rates. It would be useful to discuss how these dynamics would change with a different experimental setup or different erodibilities of the bed. I think that looking at your video footage of your experiments in depth could start to untangle some of these causes. I think it is at least important to discuss why you think these rates are so similar.

In discussion, it would be useful to compare erosional thresholds for each of your fracture orientations and spacings. Each fracture network geometry results in different block geometries, which have been shown to have a significant impact on plucking mode (see Lamb et al. 2015, Hurst et al. 2021, Lamb et al. 2009)

**References**

Hurst, A. A., Anderson, R. S., & Crimaldi, J. P. (2021). Toward entrainment thresholds in fluvial plucking. *Journal of Geophysical Research: Earth Surface*, *126*(5), e2020JF005944.

Lamb, M. P., Finnegan, N. J., Scheingross, J. S., & Sklar, L. S. (2015). New insights into the mechanics of fluvial bedrock erosion through flume experiments and theory. *Geomorphology*, *244*, 33-55.

Lamb, M. P., & Dietrich, W. E. (2009). The persistence of waterfalls in fractured rock. *Geological Society of America Bulletin*, *121*(7-8), 1123-1134.

**Esurf review aspects:**
1. Does the paper address relevant scientific questions within the scope of ESurf?
Yes
2. Does the paper present novel concepts, ideas, tools, or data?
Yes
3. Are substantial conclusions reached?
Yes
4. Are the scientific methods and assumptions valid and clearly outlined?
Yes
5. Are the results sufficient to support the interpretations and conclusions?
Not necessarily – see review.
6. Is the description of experiments and calculations sufficiently complete and precise to allow their reproduction by fellow scientists (traceability of results)?
Yes
7. Do the authors give proper credit to related work and clearly indicate their own new/original contribution?
Yes
8. Does the title clearly reflect the contents of the paper?
Yes

9. Does the abstract provide a concise and complete summary?
Yes
10. Is the overall presentation well structured and clear?
Yes
11. Is the language fluent and precise?
Yes
12. Are mathematical formulae, symbols, abbreviations, and units correctly defined and used?
Yes
13. Should any parts of the paper (text, formulae, figures, tables) be clarified, reduced, combined, or eliminated?
Yes
14. Are the number and quality of references appropriate?
Not necessarily – see other referee review.
15. Is the amount and quality of supplementary material appropriate?
Yes

---

## Author Comment (AC1)

We took the liberty to reorganise the two reviews to group similar comments. In the following, text in brown corresponds to the review by Alexander Neely (R1) and text in green is for the second reviewer (R2). Our answers are in black.

**R1: "Summary**

Fournereau et al., performed a series of flume experiments to test the role of planar discontinuities in affecting the rates and spatial patterns of erosion in a bedrock-analog material (concrete). Planar discontinuities were produced from a network of BVOH plastic that dissolves or softens in cold water. The orientation and spacing of these materials inside the concrete was systematically adjusted, providing control on the spatial distribution and behavior of material properties of the eroded bed. An abrasion mill with rotating propeller was used to drive erosion with abrasive gravels. Erosion of the bed was measured both by weighing between 2 minutes of milling and by topographic differencing using 3D scans from close-range photogrammetry. The latter approach provides a map of where and how much erosion occurred on the bed over time. These topographic scans could be shared as digital data that records a time series of how erosion progressed through the bed material. This is a nice feature and produces a lasting dataset that could then be analyzed further.

The experiments show differences in the pattern and temporal evolution of erosion that occurred between concrete only and concrete + discontinuity experiments, but the overall amount of erosion remained similar. Removal of larger particles occurred when discontinuities were present and some of these fragments remained the flume cylinder, so their particle size distribution was measured after the experiments concluded. These fragments were interpreted as "plucked" blocks, as well as other areas in the topography where hot spots of erosion occurred. The relative proportion of erosion driven by plucked blocks and presumably detachment of smaller particles was estimated and shown to be a function of discontinuity spacing and in certain cases, orientation.

The experiments are rigorous and the hypothesis is tested appropriately. My main suggestions relate to the experimental apparatus and how the results are interpreted and presented with respect to field settings, which is challenging.**"**

**R2: "**I generally agree with the review provided by the other referee on this manuscript. I think that they summarized the experimental setups and findings well. This is an interesting body of work, and I think that my main concerns lie with how the experimental set-up affects the ability to generate substantial plucking and how to relate this to a field setting. I also have concerns in the definition of plucking as used here, which could be a misconception on my part of the experimental setup. In any case, this work shows interesting relationships on how fracture spacing generates autogenic roughness in a bedrock channel. Additionally, I recommend that the authors read throughout their manuscript for spelling and grammar errors and casual writing. An example is line 326 "yet we have no clue whether it is lower""**"**

We thank the reviewers for their detailed and insightful comments, which helped us improve the clarity and relevance of our manuscript. We have revised the manuscript to better explain the limitations of our approach and to clarify the scope of our results in relation to previous experimental work on plucking and field observations. We believe these changes enhance the clarity of our study.

**R1: "Main suggestions :**

**(1) Experimental design with respect to plucking mechanism**

One potential issue with the experimental design -- combining the abrasion mill and a fractured bedrock media to quantify abrasion vs plucking erosion – is that it is not clear to what degree the mill produces flow conditions have been shown to drive plucking in fractured materials? These flow conditions were generated and isolated in experiments with different flume geometries and shown to influence plucking of blocks:

Hydraulic jumps (Wilkenson et al., 2018) – the videos in the supplement are beautiful.

Flow constrictions (Saha et al., 2021) – a very large flume underlain by blocks

Knickpoints (Dubinski & Wohl, 2013) – also a larger flume underlain by blocks

Clemence 1988 – a student paper showing plucking at a knickpoint edge.

Protrusion/angle (Sidiki Koulibaly et al., 2024) – change block orientation & joint roughness

These are just suggestions of papers. They each include literature reviews of other works that should be considered in the topic.**"**

It seems challenging to explore the significance of the non-steady flow fields in relation to plucking, given the mill apparatus design. It's difficult to interpret whether the experiment is representing these phenomena as they would be developed in a field setting. I'm curious what the authors think about digging into this point? The topography scans do exist…

**R2: "**With an abrasion mill type setup, you aren't really able to replicate the processes involved in plucking. With a purely abrasion scenario, grains circulate, eroding the bed to either produce more grains or finer material.**"**

Our setup is designed to reproduce erosion by abrasion and plucking, based on the hypothesis that a fractured bedrock submitted to sediment impact is prone to plucking. Previous studies using similar setups did not consider pre-fractured or dynamically fractured bedrock material, likely preventing plucking. On the other hand, previous experimental studies on plucking, such as the ones suggested by R1, use different setups. Most of them are done in linear channels with already detached blocks and focus on the hydraulic conditions required to induce plucking. Our setup is not designed to accurately simulate or constrain hydraulic conditions (see following comments), and we mostly focus on erosion processes induced by flow-driven sediment impact. The bedrock is not fully fractured due to the absence of a basal fracture (see following comments). In turn, this limits the ability to develop sufficient pressure fluctuations for hydraulic lift of the block. In our experiments, the hydraulic conditions allow the sediment to impact the bedrock surface with sufficient energy to 1) fracture the bedrock, in particular along the base of pre-fractured blocks, 2) lead to some slip along pre-existing vertical fracture plans, and 2) lift the block. Once a block is lifted, we observe that the shear stresses developed by the water flow are sufficient to transport the block.

The initial version of the manuscript lacked a discussion on that topic, so we added a new section (section 4.4) in the revised manuscript.

**R1: "**The experiments showed in their topography scans that more variable topography emerges depending on the fracture network. The variable topography likely generates flow features listed above as the experiments progress. The self-emergence of these features seems like a significant finding to me that is clearly shown from the experiments, and a large portion of the paper is dedicated to describing and quantifying changing patterns of erosion that result.**"**

In the initial version, we did not elaborate on the flow field and on its relationship with topography/roughness. As noted by both reviewers, this may have an important impact on sediment transport and erosion dynamics. In the revised version, we emphasize where relevant that the variable topography is a self-emergent feature that can impact the flow field (Lines: 29, 60, 540).

**R2:** "In nature, plucking is generally represented as knick progression upstream, a sort of unraveling of the bed. In addition to a fracture network of sufficient density to create blocks that are erodible under realistic hydraulic conditions, this requires a downstream boundary of an exposed block. This can occur due to vertical lift, base level drop, extreme abrasion, etc. In the abrasion mill setup, you don't have the ability to initiate the plucking with a downstream boundary (at least as you have set it up with a flat bed). Which in and of itself is okay, because you just have to simulate the 'first' plucking event by lift to trigger plucking. But then the amount of erosion by plucking that you are able to generate is only one circumference of your circle until you reach the first block again. Then you must re-create the higher erosional threshold first plucking event to trigger a wave around your disk.."

Indeed, plucking needs some "initiation point". In our experiments, the initial topography is quite flat with a bit of roughness but as shown on Figure 4, we observe the removal of blocks quite early in the experiments. This implies that the sediment impacts and flow conditions are sufficient in our experiment to induce plucking without a base level drop or extreme abrasion. As noted in the previous comment, we suggest that the sediment impacts are able to break the bedrock in depth, forming a fully pre-fractured block that can be lifted either by the flow or by another impact.

Because plucking emerges without initial "seed" or initial exposed block, it can occur at multiple locations at the same time (see Figure 2f, 3a or 6a). In addition, in our experiments, we do not observe a proper "bed unravelling" as often observed in previous experimental studies (see previous comment) yet we do observe spatio-temporal trends (ie, succession of plucking events next to each other through time). However, and we believe this is very interesting, plucking propagates either downward or upward. We suggest that this reveals the key role of sediment impact on plucking when blocks are not fully detached. Once a block is plucked, this creates local roughness and bed exposure that favors sediment impact, enhancing in turn the likelihood of fully detaching a block. Once this is done, the block can easily be removed by the flow. Therefore, the limiting factor in our setup is the capacity to break blocks, not the capacity to lift them. Finally, as shown in Figure 4a, the removal of one or several blocks at the same location does not imply that the full circumference is going to erode by plucking. Again, we think this reveals the key role of sediment impacts. In the revised version, we added a new paragraph (section 4.4) to address this point that was missing in the initial version.

**R2:** "Further, in the real world these larger blocks would wash downstream and exit the system (or deposit downstream). What happens to them in your mill? This is not necessarily unrealistic, but deserves discussion."

We forgot to mention the fate of the detached blocks in the initial version, thanks to the reviewer for pointing at it. When the blocks are detached and entrained, they are mixed with the granite grains used as impactors and become part of the bedload. However, concrete blocks are lighter and weaker than the granite grains so that they tend to be eroded down to sand. Moreover, when we empty the column every two minutes (see Experimental protocol), we remove all the concrete chunks that we can find. We thus work at almost constant mass of sediment. We therefore make the hypothesis that the concrete blocks do not contribute significantly to erosion while in transport.  This is a bit different

to natural settings, where in-situ sediments become part of the bedload and contribute to erosion downstream. This is now clearly stated in the revised version. (Line: 177)

**R2:** "I think also this hinges on your blocks being fully separated from the bed and readily mobilized. If you don't have a 3D fracture network, then are these events really plucking?"

Our concrete disks are composed of two different materials (concrete and plastics) with different density, mechanical property and erodibility, but the whole disk remain cohesive. As clarified above, the fracture networks are 3D as they form a 3D connected network, yet, the "bottom" fracture has to form in situ during the experiment (i.e, the blocks are not fully detached and their thickness is an emergent property) (Lines: 107, 147, 220, 347, 491). Once the bottom fracture has ruptured the entire base of the block, a block can be entrained thanks to sediment impact and/or hydraulic lift. This process corresponds to plucking. Please note that the size of the blocks is not systematically controlled by the geometry of the fracture network (Figure 4 a; Lines: 291, 320, 492). In the revised version, we rephrased the end of the introduction to avoid any confusion.

**R2:** "Or are sediment impacts able to abrade larger chunks of the cement between fractures?"

In the introduction (l. 94), we discuss the continuum in erosion processes, from abrasion to plucking via macro-abrasion. As far as we know, there is no clear limit between these 3 processes that can occur at any time in a bedrock river (see references in the Introduction). We believe that the 3 processes occur in our experiments, however, we don't have the time and space resolution to distinguish, considering that it would be possible, between abrasion and macro-abrasion. Therefore, as stated in line 110, we focus on abrasion and plucking only and use geometrical thresholds (area and depth) to separate the two (see section 3.3). We believe that the revision of the manuscript will help make this point clearer.

**R2:** "It would be interesting to run these experiments with a range of sediment inputs. I'm curious if with your setup you would even get any 'plucking' if you don't have sediment impacts since it seems you don't have fractures at the base. The total percentage of erosion by plucking will depend on how abradable your bed is compared to how pluckable your blocks are."

As the disk is fractured but not broken, we believe the blocks are not pluckable from the beginning of the experiment. We expect that without sediment impacts, the bedrock will not be able to be plucked in our experiments. In the work presented here, we use a constant mass and size of sediments to ensure similar conditions of erosion. We fully agree that it will be interesting to explore the role of the sediment input. However, it is beyond the scope of this paper and out of timing for the first author, Marion Fournereau, who will finish her PhD in a few months. But yes, as already mentioned in the manuscript (section 4.2), in the future, we wish to run new experiments in which we vary the size of the sediments while keeping the mass constant (to have the same total energy of impact, Sklar and Dietrich, 2009).

**R2:** "In discussion and throughout paper you need to be clear that you are exploring a limited scenario where you have a constant supply of sediment. This interaction of sediment can really impact the dynamics of plucking vs. abrasion."

In addition to the initial mentions (Introduction and Methods), we modified the text where relevant. (Lines: 186, 519)

**R1:** "Fractured bedrock tends towards these channel geometries (constrictions & knickpoints) in field settings due to heterogeneity in the rock strength fabric. Not sure if a reference is needed here, but a 600 m long scan of a bedrock river section in Taiwan shows channel widths narrowing from >100m to 40m over a distance of a few 10s of meters for example (Carr et al., 2023)."

Thanks for pointing out this reference. We now discuss this paper by Carr et al. (2023) in the new paragraph dedicated to the links between our study and natural systems (section 4.5).

Carr, J. C., DiBiase, R. A., Yeh E., Fisher D. M., Kirby E. : Rock properties and sediment caliber govern bedrock river morphology across the Taiwan Central Range. Science advances, 9(46), eadg6794. https://doi.org/10.1126/sciadv.adg6794, 2023.

**R1:** "The types of conclusions that can be drawn from the work are maybe a bit limited when applying to field settings if the above paragraph can't be quantified – namely it's tough to conclude a lack of increasing erosion effectiveness from adding fractures in a field setting? The self-emergence of a rougher bed seems like a significant point that could maybe be expanded on."

It is indeed difficult to apply outcomes obtained from experimental works to field settings. Still, some interesting comparisons can be drawn. We have therefore added a new section (section 4.5) in the revised manuscript to discuss the relevance of our experimental outcomes for field settings.

**R1: " (2) Experimental design with respect to the 3D connectivity of the joint network**

The pre-seeded fractures do not contain an interconnected 3D network of joints. There are no subhorizontal fractures if I am understanding correctly? So blocks are not readily available for plucking – the 3rd joint set needs to form in-situ."

**R2:** "I had a hard time understanding the fracture network within your discs. Were the fracture networks 3D, or did you just have 2D fractures? I think that a conceptual figure for both this and how you sum up dip angles and fracture lengths would be really helpful."

Our fracture networks are indeed composed by two sets of vertical fractures, as a 3D mesh of 2 cm high. Indeed, the 3rd joint needs to form in-situ to make the block available for plucking as we never observe a 2-cm high block. We rephrase the text where relevant. The geometry of the fractures is given on Figure 1a and the fracture length is calculated in map view before any erosion occurs. We made this explicit in the revised version. (Lines: 107, 147, 220, 347, 491)

**R2:** "What are dimensions of blocks created by your fracture network? Block geometry has been shown to have a significant impact on plucking mode and erodibility (see Lamb et al., 2015; Hurst et al., 2021; Lamb and Dietrich, 2009), and so the dimensions of blocks created by your fracture network should affect how 'pluckable' the blocks are."

The fracture network represents preferential zones of weakness, which favor the formation of pluckable blocks but do not impose it. The geometry of the fractures, which represents preferential zones of weakness, is shown and explained in different parts of the text, in particular in section 3.3. The geometry and scaling laws describing the plucked blocks are described in Fig. 7 and S2 and in

section 4.2. The results presented here demonstrate that indeed, the dimensions of the blocks are influenced, but not prescribed, by the spacing of the fracture network.

**R2:** "In discussion, it would be useful to compare erosional thresholds for each of your fracture orientations and spacings. Each fracture network geometry results in different block geometries, which have been shown to have a significant impact on plucking mode (see Lamb et al. 2015, Hurst et al. 2021, Lamb et al. 2009)"

In all our experiments, we use the same amount of sediment and the same flow velocity so that the shear stress imposed on the disk is the same. Determining the minimum shear stress to exert to observe plucking according to the fracture network would require a totally different setup, or strong hypothesis that would limit the impact of such approach. Therefore, we do not explore this interesting question in the revised manuscript.

**R1:** "It is hard to quantify how the fracture network develops/grows in near surface stress fields, particularly quantifying the openness of 3D joint networks in valley bottoms, because the horizontal fractures would be mostly buried. Fracture surveys on pinnacles, spires, and waterfall edges seem to indicate that blocks are detached by 3D joint sets with open apertures though.

Flow underneath the blocks along a subhorizontal fracture set is important for generating lift or hydraulic jacking from sediments infiltrating cracks (Hancock et al., 1998). The latter mechanism would not be possible in this experiment given the aperture of the fractures (~1 mm) and the size of the sediment used (10-20 mm). This demonstrates scaling issues between the flume experiment design and particle size distributions, which could be significant for driving erosion in natural streams. In the field, the fracture aperture can be wider than a portion of the particle size distribution."

We agree with this comment, and in our experiments, there is no fracture aperture as they are "filled" with the plastic material. Therefore, we expect plucking to be potentially more efficient in natural settings (due to hydraulic pressure in the horizontal fracture and to fine material infiltrating the cracks) than in our experiments. In the revised version, we address this point in the new section 4.5 dedicated to the links between our experiments and field settings.

**R1: "(3) Strength of the joints**

I am not sure how to interpret the material used to form the joints, but I do find it very interesting. Is the material harder or softer than the concrete, and how does it change over time? Could this be calibrated with submersion experiments without any sediment? In either case, there are landscape examples where joint material is softer (as in an aperture) or harder than the surrounding rock mass (joints infilled with carbonate cement or vein material – qtz for example), so there is significance in putting in the heterogeneous material with a pre-defined orientation & spacing in either direction."

It is very difficult to assess the relative properties of the two materials once they are mixed and immersed. Based on observations and on their mechanical properties given by the producer or determined in the lab (section 2.1), we make the hypothesis that the plastic has potentially a larger tensile strength than the concrete, which is often put forward as a prime mechanical control of erodibility. This could explain why the disks with a low fracture density erode slower than the concrete disks without any plastic (section 3.2). Our fractures therefore introduce a contrast in mechanical resistance, which is not well constrained. This is a limit of our setup, as we can imagine that with a different plastic, the plucking dynamics would be different. We now address this point in section 4.4.

**R1:** "These complexities (1-3) may limit the scope of interpretations that can be made from the experiments when moving to natural landscapes/channels, but the research question addressed by the experiment is very difficult. The experiment is performed rigorously and with reliable measurements of erosion patterns. The experiment setup may apply to certain field settings and the general observation of more spatially variable/rougher bed topography created from the joints is important."

Thanks to the reviews, we now better discuss the links with natural systems by adding a dedicated paragraph, bearing in mind that experiments don't seek to reproduce a specific natural system but explore the dynamics of processes. We also emphasize where relevant the impact of plucking on roughness and bed variability. (Lines: 29, 60, see new section 4.5)

**R2:** "You also have a highly erodible bed that is susceptible to abrasion, which can influence the dominance of abrasion over plucking in terms of overall erosion. "

We fully agree that our results would be different with a different disk rheology. However, the bedrock material used in our experiments  was not designed to simulate a specific substrate but rather to obtain a material erodible enough, to results in significant erosion over a reasonable experimental time (a few cm per hour), and yet coherent enough to enable both plucking and abrasion. All experiments were done with the same mix so that their erosional behaviour (abrasion vs plucking) can be compared and discussed. Where relevant, we always mention that the results are with respect to our setup and material (Lines: 211, 315). We added a section to address links with natural systems (section 4.5) in the revised version to make it even clearer.

**R2:** "To me, the fact that there is so little difference in average erosion rates indicates one of two things. Either:

a) the experimental setup is preventing a greater magnitude of plucking from occurring and since the bed can't unravel, the stochastic plucking events are contributing a great magnitude of erosion that is averaged out over time (a real thing that happens where rapid events with high magnitudes of erosion are interspersed with long periods of no erosion!!! i.e. jokulhaups in Iceland. So a cool interpretation if you can back it up with observations or data) or

b) erosion by abrasion dominates the long-term erosion rates. It would be useful to discuss how these dynamics would change with a different experimental setup or different erodibilities of the bed."

As answered to the previous comment, our experimental setup enables both plucking and abrasion, and changing the rheology of the concrete and/or the plastic could result in a different behaviour. We believe the tensile strength of the experimental fracture network might be relatively significant compared to the one of the concrete. In turn, our experiments are likely favouring abrasion over plucking. Yet, to fully answer this question, we would need to run a new series of experiments where rheology, fracture density and sediment size would be systematically varied allow plucking to occur and how this relates to natural systems. However, this is beyond the scope of this study. In the section 4.4 of the revised version, we now mention how rheology could affect our results and we discuss our results with respect to previous works on plucking that were mostly done in linear channels with readily pluckable blocks.

**R1:** "I'd recommend if possible that the authors include some pictures of the bed as the experiments progress? My understanding is these abrasion mills turn into a mess when they are running, but some photographs of the disks (if fractures formed in-situ) or wear of the BVOH might be insightful over time. Many flume studies include photographs or videos of the experiments progression, and I find this the be a big advantage to these experimental set ups."

**R2:** "I think that looking at your video footage of your experiments in depth could start to untangle some of these causes."

We agree that it would be super interesting to follow in real time what is happening in the mill. We thought about different options but none of them was satisfying because, as noted by R1, it is a mess during erosion (water becomes turbid due to small particle, so side view is very limited, and the propeller prevents any view from above). Pictures of the disk from above after the sediments and water removal are not very informative either as it is difficult to see any topography with such point of view. This is why we choose to present only the 3D topography and erosion maps, and for the sake of concision, we focus on only 3 experiments on a few time steps (Figure 2). We believe this is enough for the main manuscript but to give a better idea of the dynamics, we now provide the full time series (i.e., with an image every 2 minutes) for the topography and the erosion rate, as Supplementary Material (Figure S5).

**R1: "Line by line comments**

Lines 24-28 – any quantitative comparisons/statistics to report? In some of these lines (line 28) its not so clear whether this conclusion is based on the experiments or a general statement."

We agree that the initial version was ambiguous, so we rephrase to make it clear that these lines are based on our experimental results only (Lines: 26-27).

R1: "Line 29: Here is a place where it could be mentioned that erosion mode and location affects the roughness of the bed and presumably the flow field? Just a suggestion."

Thanks for pointing this potential implication of our work! We added a dedicated sentence to the abstract. (Lines: 29)

**R1: "Line 67-80** – In the context of fluvial erosion and bedrock fractures, there are fracture density measurements recorded in Snyder et al., 2003. I'll throw in a quick blurb here, but it's maybe not too important. In some ways DiBiase et al., 2018 or Neely & DiBiase, 2023 are making these comparisons between channel geometry, erosion, and fracture spacing, but the difficulty is that sediment size and bedrock fracture spacing co-vary in those sites (data from Neely & DiBiase, 2020), so the effect of one or the other is difficult to isolate – we interpreted that channels are mantled with sediment, so the fracture spacing effect is secondary to the cover effect. The flume approach is nice in this aspect -- the experiment can control the ratio between fracture spacing and sediment size in ways that might not be possible with field studies."

We also believe that the experimental approach is a nice way to overcome some limitations inherent to field work, like mentioned by the reviewer. In the revised manuscript, we better discuss the relation between field and experimental studies, in particular in the new dedicated section 4.5.

**R1:** "Line 81 – should plucking be included here? If the bedrock is already sectioned into blocks?"

Indeed, mention to plucking was missing in this sentence. This is corrected in the revised version (line 94)

**R1:** "Fig S2: interesting comparison, is the SfM of cumulative erosion always compared to the topography of the first point cloud? Or is erosion summed between successive differences as erosion proceeds? Just wondering if that could be a reason for the "drift" in cumulative erosion rates?"

For both methods, the cumulative erosion is calculated relative to the first weighting or initial cloud (this is explicitly stated in the revised version). The erosion rate is determined from consecutive clouds. The difference between the curves might be due to different sources of uncertainties according to the method. The weighting method seems affected by two processes: 1) the disks are saturated in water before we start the experiments, yet, we often observe an increase in weight during the first 10 minutes. We believe this is due to further infiltration of water once the surface starts to be eroded. 2) We sometimes observe sand and water below the disk that we are not able to remove before weighting. Both processes induce some variations that do not influence the 3D data. On the other hand, 3D data rely on the quality of the pictures and point cloud reconstruction. This is why we use the weighting as an (unperfect) control. Despite these differences, in most experiments, the curves follow the same trend, in particular if we focus on the erosion rates. As the manuscript is mostly based on erosion rate and used the 3D data only, the difference between the two methods does not affect the results.

**R1:** "Figure 3 is very informative. I like panel d a lot. So if am reading this correctly, the largest local erosion rate is ~5 mm/min. Does this correspond to the size of a plucked block? Or are the blocks mostly wearing in place, in between the fractures? I guess with 10-20 mm impactors and 10-20 mm blocks, the impactors may not really be able to penetrate between the fractures"

As stated in 2.3, the local erosion rate is the erosion observed at each point of the raster for each time interval. Such a high rate could be due to the removal of a large block or to the removal of several blocks at the same place during the 2 min time interval. On Figure 7a, we show the relationship between the pluck depth and the pluck area. This is not strictly equivalent to Figure 3d, but it suggests that large erosion local rates are associated with areas ranging between 100 and 1000 mm$^2$, ie, sometimes more than the area of a single block defined by fractures.

**R1:** "Figure 4: Could it be more difficult to detect the small plucking events in the highest fracture density case?"

**R2:** "Why do you think less plucking in the highest fracture density? Are the plucked blocks so small that they are below your threshold of detectability for plucking?"

As explained at the beginning of section 3.3, we define plucking from two thresholds based on visual observations and on Figure 3. So strictly speaking, there is no bias in plucking detection, whatever the network. In addition, plucking events tend to be small and depth of plucking is also limited when the network is dense (see Figure 7) yet, not smaller than in other experiments. So, we are confident that the results show on Figure 4 are robust and not due to a detection issue. Yet, we acknowledge that a different definition of plucking would lead to slightly different graphs for all experiments.

**R1:** "I have a hard time understanding total dip angles? So two joint sets at 45 degrees would be 90 degrees?"

Yes, this is stated in section 2.1 and values are given in Table 1. We agree that this proxy is not standard, but we did not find any better in the literature. We tried different manners to discriminate in a simple way between these experiments and the sum of the dip appears as the most synthetic and informative. We suggest in section 3.4 that this sum describes somehow the difficulty to remove large and deep blocks at once.

**R1:** "Figure 5c, if more detail is needed to distinguish joint set orientations – i.e. same angle or different angle, consider using different symbols for the markers? Figure 6 – I still have a bit of a hard time understanding the geometry by summing the dips together, maybe there is another way to display this. I think simply treating each combination of dips as a different column could be informative. Information about the fracture geometries gets lost by summing the fracture set dips together."

We thought about using different symbols to indicate networks with the same or different angles (ie, 4545 vs 4567). However, this characteristic does not seem to exert a major control on erosion rate (Figure 5) so we simplified the symbols by using only color and kept the same symbol for Figure 5. The relationship between the sum of dip and the geometry of network, together with the colorcode, are given in Table 1. We believe this is enough for the reader to understand the main purpose of this already a bit complex paper.

**R2:** "Fig 5/dip angle variability conclusions. How many runs did you have at each dip angle? It looks like you had greater variability in runs where you conducted more repetitions. So that isn't a strong conclusion if you only had 1-2 runs in the lower variability cases and would need further discussion and analysis."

The number of runs is given in section 2.1 and the characteristics of each experiment are given in Table 1. At the beginning of section 3.4, we indicate that we have 7 experiments to explore the role of the dip angle. We replicate only experiments with symmetrical networks (see Table 1). We acknowledge that this is a limited data set, yet, this is the first to explore this question, so we assume that it is worth of interest as it is.

**R2:** "Due to the circular nature of your disk, the fracture orientation relative to flow is never consistent so it is difficult to draw conclusions about fracture orientation and dip angle. These orientations relative to the flow direction are extremely important, and having them change is likely a cause of the variability in your data."

Indeed, the fracture orientation with respect to the flow is never the same as we use a circular flume. As a result, when the fracture dip is not vertical, the fractures tend to be aligned with the flow on one part of the disk and more facing the flow on the other side. This induces some very interesting patterns presented in section 3.4 and as far as we know, this is the first exploration of this process. Once again, we acknowledge that our data set is a bit limited and that experiments have intrinsic limitations (see section 4.1), yet, we are confident that these results are of interest for the community.

**R2:** "Line 254-255 "and we therefore suggest that the shape of the local erosion rate distribution informs on the occurrence of plucking." Considering that plucking contributes to the erosion rate

distribution, the erosion rate distribution cannot control the occurrence of plucking. Maybe this needs to be reworded for clarity."

**R1:** "Line 319: I'm not sure, there are quite a bit of experiments that include fractures in the form of pluckable blocks? – maybe rephrase to be more specific about what is different about your experiment than some of the experiments listed earlier in the review."

In fact, previous studies already worked on plucking but as you mention, the blocks were already pluckable whereas in our experimental setup, the 3rd join has to form in situ. We wanted to stress this difference, but the sentence was indeed unclear. It has been rephrased in the revised version.

**R1:** "Line 326: I really like the experiments, but it would be great to add some clarity here. Perhaps you can calibrate the material properties by submerging for regular intervals and measuring?"

It would be great to do so however, we could not find a way to do it in a proper manner (see answer to previous similar comment)

**R1:** "Section 4.2 – I feel like something got disorganize here when explaining the methods of defining a "plucking event". Could this be elaborated on in the methods section 2.3? So moving part of section 3.3 for the methods into section 2.3?"

In the Introduction, we now better state how we define plucking, yet, the thresholds that we use to define plucking are derived from experimental results (see Figure 3) so that we can't move the mentioned part of section 3.3 earlier. (Lines: 66, 107, 216)

**R1:** "It seems unusual to me that a plucking event could be defined as eroding material that is smaller than the fracture-bound blocks, but this is maybe an outcome of not having a sub-horizontal fracture set as well?"

When blocks are readily pluckable, their removal might be due to hydraulic forces and/or to sediment impact delivering some energy. In our experiments, a block must fully form before it can be removed. This is why we define plucking from geometrical thresholds (area and depth) rather than from the geometry of the fracture network. As you mention, some blocks are smaller than the area defined by the fractures and we also observe (and recover after the 2 min time interval) a few blocks with a mix of plastic and concrete. This suggests that plucking from fractured bedrock is a bit more complex than plucking from pluckable blocks, but this is beyond the scope of this study to investigate this process further. In the revised manuscript, we make more explicit that the size of the blocks does not always correlate with the fracture spacing and mention that it is an important difference with previous experimental works focused on plucking.

**R1:** "I see the fragment sizes plotted as a figure in the supplement. This is pretty interesting and does show more "platey" fragment sizes. Do the fragments cluster around the fracture spacing dimensions? I would expect the clasts corresponding to the densest fracture density to produce a cluster of fragments near 10mm a-axis, 10 mm b-axis, if the partings were occurring along the fractures? I guess I am trying to untangle what went on inside the flume during detachment. The fragments are helpful but as you say they are an indirect measure. Glad you measured and held onto them!"

Thanks. The fragments are indicative, but we cannot determine if they were eroded or broken after removal. This is why we kept this figure as a supplement and did not go too far with these data.

**R1:** "Figure 7: Interesting plots that are getting at some of the geometry questions from before. So 10x10 mm is the spacing between the densest fracture network, but can produce plucked blocks that have a z-axis between 2.2 and 4 mm? The units of this are a bit hard to follow with the dimensionality of a and d, and the fit power law form. Maybe plot without log transformation? If you are producing cubes, the relationship should be a square root form? So this is implying that slabs are formed (makes sense for plucking)?"

This figure simply shows the relationship between the area (in mm$^2$) and the depth (in mm) of plucking events (depth being defined as the volume divided by the area, see beginning of the section). The log transform is relevant to enhance visualization as the plucked depth ranges over less than an order of magnitude while the plucked area spans about two. The color code gives extra information with the fracture density on panel a and the area between fractures on panel b. This area between fractures could be seen as the unitary area of plucking blocks form only along fractures. However, we discussed above that this is not always the case. This explains that we observe a continuum in plucked areas instead of discrete values. As a result, we do not produce cubes but rather parallelepipeds. And yes, we do observe plucking events with a thickness of a few millimeters in experiments with dense (10x10 mm) network. We also found a few blocks that confirm this observation (figure S2). In the revised version, we removed the fit as we found it induces some confusion (units were not consistent) without adding a lot of information (the relationship is easy to observe).

**R1:** "there is limited discussion linking the results to past work on the topic – either other flume experiments or contextualizing the results with bedrock erosion in the field. Maybe this can be developed a bit more? It is important to discuss the data and the specifics of the experiment & design, but a section that zooms out to contextualize the study with past work/field settings would be encouraged (by me)"

**R2:** "While your discussion section does a nice job of talking about all of your experimental runs together, I think that more needs to be done to put your work into the context of previous work and discuss how this applies to natural settings."

We fully agree that the initial version lacked such a discussion, so we added two dedicated sections in the Discussion (new sections 4.4 and 4.5) and modify the text where relevant to better acknowledge similarities and differences with previous experimental works. (Lines: 57, 86)

We addressed all of the minor comments where relevant in the revised version of the manuscript.

**R1:** "Line 13: hillslope dynamics? Maybe "incision at the base of hillslopes" – if you have enough words to spare

Line 16: "However, there is no systematic study of the impact of fracture geometry and density on bedrock river erosion" Slight wording changes suggested because this could be misinterpreted. Maybe try:

*"Systematic studies comparing fracture geometry and density and bedrock river erosion are needed."*

Line 22: I'd remove "to our knowledge, this is the first study of its kind" --- if it's meant to shine, the work will shine on its own

Line 40 – I like this point, but I would reword. Flume experiments also have their challenges upscaling to landscapes (which is usually a goal).

*"Experimental approaches can be an attractive approach to address these questions due to slow timescales of erosion and numerous conflating factors that could affect bedrock strength in field settings."*

Line 42: "demonstrated" Line 43: "maximized" Line 45: new paragraph?

Line 75 and others: unfortunately, the plural of "bedrock" is "bedrock".

Figure 1 – I like this figure and it is nice to plot the control run in the context of the other studies with similar set ups.

Line 128: "center"

Line 170 – extra period.

Line 175 – interesting, this is a good idea.

Line 181, 185; in a few places in the manuscript would change "weighted" to "weighed"

Figure 2 – Interesting to see the different spatial patterns in erosion and the emergence of the plucking holes

Line 199: "fractures"

Line 206: "eroded less" instead of "barely"

Line 212: in a propellor flume – not sure if there is a better word to describe the flume set up.

Line 304: "Asymmetric"

Line 328: "protruding from"

Line 331: "resistant"

Line 347: organization

Line 353: the geometries you are using seem like a good start to me?

Line 363: "implies"

Line 370: "ones?" fragments?

Line 436: remove "the" before erosion rates

Line 437: "no fractures"

Line 442: I like this sentence for the implications of the work.

Line 446: remove "or not"

Line 448-449: I would be careful about the last phrase, unless addressing the experimental design/questions earlier in the review.

Thank you for sharing your interesting work and it was a pleasure to read. If you would like to discuss more or if you have any questions about this review, please do not hesitate to contact (abn5031@arizona.edu)"

**Referenced studies:**

Carr, J.C., DiBiase, R.A., Yeh, E.C., Fisher, D.M. and Kirby, E., 2023. Rock properties and sediment caliber govern bedrock river morphology across the Taiwan Central Range. *Science Advances*, *9*(46), p.eadg6794. https://doi.org/10.1126/sciadv.adg6794

Clemence, K.T., 1988. 1987 Student Professional Paper: Undergraduate Division: Influence of Stratigraphy and Structure on Knickpoint Erosion. *Bulletin of the Association of Engineering Geologists*, *25*(1), pp.11-15. https://doi.org/10.2113/gseegeosci.xxv.1.11

Dubinski, I.M. and Wohl, E., 2013. Relationships between block quarrying, bed shear stress, and stream power: A physical model of block quarrying of a jointed bedrock channel. *Geomorphology*, *180*, pp.66-81. https://doi.org/10.1016/j.geomorph.2012.09.007

Hancock, G.S., Anderson, R.S., and Whipple, K.X., 1998, Beyond power: Bedrock river incision process and form: in Tinkler, K., and Wohl, E. E., eds., Rivers over rock: Fluvial processes in bedrock channels : Washington, D.C., American Geophysical Union, Geophysical Monograph 107, p. 35-60.

Koulibaly, A.S., Saeidi, A., Rouleau, A. and Quirion, M., 2024. Laboratory physical model for studying hydraulic erodibility of fractured rock mass. In *New Challenges in Rock Mechanics and Rock Engineering* (pp. 1416-1422). CRC Press.

Saha, R., Lee, J.S. and Hong, S.H., 2021. The impact of lateral flow contraction on the rock plucking process under sub-critical flow conditions. *Earth Surface Processes and Landforms*, *46*(14), pp.2902-2915. https://doi.org/10.1002/esp.5220

Snyder, N.P., Whipple, K.X., Tucker, G.E. and Merritts, D.J., 2003. Channel response to tectonic forcing: field analysis of stream morphology and hydrology in the Mendocino triple junction region, northern California. *Geomorphology*, *53*(1-2), pp.97-127. https://doi.org/10.1016/S0169-555X(02)00349-5

Wilkinson, C., Harbor, D.J., Helgans, E. and Kuehner, J.P., 2018. Plucking phenomena in nonuniform flow. *Geosphere*, *14*(5), pp.2157-2170. https://doi.org/10.1130/GES01623.1

Hurst, A. A., Anderson, R. S., & Crimaldi, J. P. (2021). Toward entrainment thresholds in fluvial plucking. *Journal of Geophysical Research: Earth Surface*, *126*(5), e2020JF005944.

Lamb, M. P., Finnegan, N. J., Scheingross, J. S., & Sklar, L. S. (2015). New insights into the mechanics of fluvial bedrock erosion through flume experiments and theory. *Geomorphology*, *244*, 33-55.

Lamb, M. P., & Dietrich, W. E. (2009). The persistence of waterfalls in fractured rock. Geological Society of America Bulletin, 121(7-8), 1123-1134.